# ByteMorph: Benchmarking Instruction-Guided Image Editing with Non-Rigid Motions

## Abstract

Editing images with instructions to reflect non-rigid motions—camera viewpoint shifts, object deformations, human articulations, and complex interactions—poses a challenging yet underexplored problem in computer vision. Existing approaches and datasets predominantly focus on static scenes or rigid transformations, limiting their capacity to handle expressive edits involving dynamic motion. To address this gap, we introduce ByteMorph, a comprehensive framework for instruction-based image editing with an emphasis on non-rigid motions. ByteMorph comprises a large-scale dataset, ByteMorph-6M, and a strong baseline model built upon the Diffusion Transformer (DiT), named ByteMorpher. ByteMorph-6M includes over 6 million high-resolution image editing pairs for training, along with a carefully curated evaluation benchmark ByteMorph-Bench. Both capture a wide variety of non-rigid motion types across diverse environments, human figures, and object categories. The dataset is constructed using motion-guided data generation, layered compositing techniques, and automated captioning to ensure diversity, realism, and semantic coherence. We further conduct a comprehensive evaluation of recent instruction-based image editing methods from both academic and commercial domains. The benchmark is available here.

## 1 Introduction

The rapid advancement of multi-modal datasets Lin et al. (2014); Schuhmann et al. (2022) and generative modeling techniques Goodfellow et al. (2014); Dinh et al. (2014) has substantially enhanced the capabilities of text-to-image (T2I) generation models Rombach et al. (2022); Saharia et al. (2022); Yu et al. (2023); Yang et al. (2024a). Building upon these advancements, instruction-based image editing methods Brooks et al. (2023); Zhao et al. (2024); Zhang et al. (2024); Sheynin et al. (2024); Zhang et al. (2023b) have emerged as powerful tools that enable intuitive visual modifications through natural language instructions without necessitating explicit masks or annotations. Despite this progress, existing instruction-guided editing techniques primarily address static or localized edits, and largely overlook the complexities associated with non-rigid motion edits, such as dynamic camera movements, deformable object transformations, human articulation, and interactions between humans and objects. While recent works have contributed datasets tailored for instruction-following editing Brooks et al. (2023); Zhang et al. (2024); Fu et al. (2023); Zhang et al. (2024); Zhao et al. (2024); Sheynin et al. (2024); Yu et al. (2024), they predominantly focus on appearance-centric alterations and fail to adequately capture dynamic spatial relations and broader scene transformations. Consequently, these limitations prevent models from acquiring the nuanced, motion-oriented editing capabilities necessary for realistic and expressive manipulation of visual content.

To bridge this significant gap, we propose ByteMorph, a comprehensive framework designed explicitly for instruction-based image editing focusing on non-rigid motions. Central to ByteMorph is ByteMorph-6M, a large-scale benchmark dataset comprising diverse and high-quality image editing examples that explicitly incorporate dynamic transformations from video, including camera movements, pose adjustments, object dynamics, and scene evolutions driven by interactions. Each data instance in ByteMorph-6M consists of a source image, a motion-focused natural language instruction, the corresponding edited image, and descriptive captions for both source and edited images. To build the dataset, we first generate diverse and expressive videos using an image-to-video model. We then extract motion-aware frames and pair them with automatically generated instructions that describe the intended transformations. This pipeline ensures that the generated data is both semantically rich and

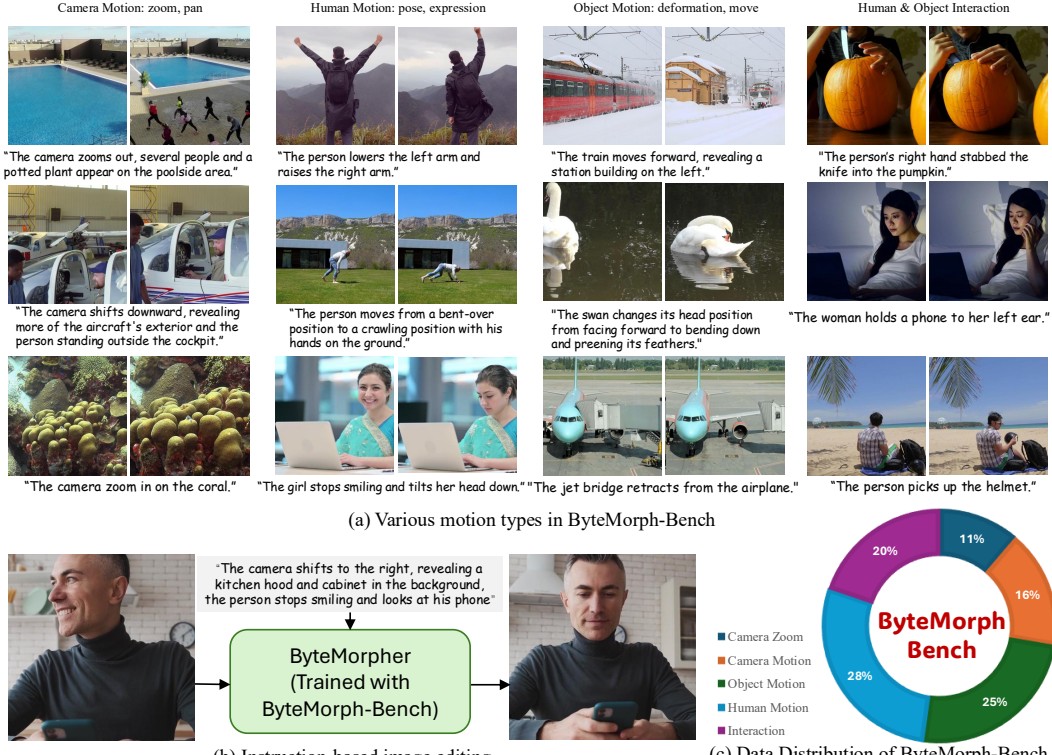

(a) Various motion types in ByteMorph-Bench

(b) Instruction-based image editing

(c) Data Distribution of ByteMorph-Bench

Figure 1: **Overview of ByteMorph**. a) We first construct ByteMorph-6M and ByteMorph-Bench, a large-scale dataset and a corresponding evaluation benchmark with data that covers a diverse range of non-rigid motions. b) We then fine-tune ByteMorpher, a diffusion transformer model initialized with pretrained weights from .1-dev Black Forest Labs (2024), using the data collected in ByteMorph-6M. The distribution of ByteMorph-6M is shown in c).

visually realistic. For rigorous assessment, we additionally provide a curated evaluation benchmark, ByteMorph-Bench, containing 613 challenging test samples spanning various categories of non-rigid motion edits.

Leveraging ByteMorph-6M, we further introduce ByteMorpher, a Diffusion Transformer baseline model specifically tailored for motion-aware editing tasks. ByteMorpher integrates multi-modal attention mechanisms, effectively capturing complex non-rigid transformations guided by natural language instructions. Through extensive evaluations across multiple dynamic editing scenarios, we demonstrate that ByteMorpher substantially surpasses current open-source methods Tan et al. (2024); Cao et al. (2024); Zhao et al. (2024); Brooks et al. (2023); Zhang et al. (2024); Yu et al. (2024), particularly excelling in edits involving viewpoint adjustments, articulated movements, and multi-object interactions. Our experiments also highlight the challenges of existing foundation models from the industry StepFun (2025); Google (2025); OpenAI (2025); ByteDance (2024); vivago.ai (2025); Google (2025) in preserving realism and accurately following instructions within dynamic editing contexts, thereby emphasizing the necessity for specialized datasets such as ByteMorph-6M and further research for building advanced models upon ByteMorpher.

Our contributions are summarized as follows:

- We introduce ByteMorph, a unified framework for expressive and instruction-based image editing encompassing non-rigid motions.

- We present ByteMorph-6M and ByteMorph-Bench, a large-scale dataset and a comprehensive benchmark, with high-quality image pairs for training and evaluation, addressing various dynamic editing scenarios, including camera motion, object transformation, human articulation, and human-object interaction.

- With the proposed dataset, we finetune ByteMorpher, a DiT-based model specifically developed for instruction-guided, motion-centric image editing, setting a baseline for performance on non-rigid motion editing tasks.

| Dataset | Size | Non-Rigid Edit | | | | Other Edits) | Image Pair Generation | |
| --- | --- | --- | --- | --- | --- | --- | --- | --- |
| | | Human Pose | Object Deform. | H-O Interact. | Camera Motion | (Stylize, Attr. Mod.) | Sequential (From Video) | Model-Edited (From Images) |
| MagicBrush Zhang et al. (2024) | 10K | × | × | × | × | ✓ | × | ✓ |
| InstructPix2Pix Brooks et al. (2023) | 313K | × | × | × | × | ✓ | × | ✓ |
| HQ-Edit Hui et al. (2024) | 197K | × | × | × | × | ✓ | × | ✓ |
| EditWorld Yang et al. (2024b) | 8.6K | × | ✓ | × | × | ✓ | × | ✓ |
| UltraEdit Zhao et al. (2024) | 4M | × | × | × | × | ✓ | × | ✓ |
| AnyEdit Yu et al. (2024) | 2.5M | ✓ | ✓ | × | ✓ | ✓ | × | ✓ |
| ByteMorph-6M | **6.4M** | ✓ | ✓ | ✓ | ✓ | × | ✓ | × |

Table 1: Comparative analysis of popular publicly available instruction-based image editing datasets. ByteMorph-6M has a dedicated **focus on non-rigid image editing**. Its image pairs are uniquely **derived sequentially from video data**, ensuring natural content consistency and realistic transformations, ideal for complex motion editing tasks. It provides extensive coverage (6.4M total images including Human Pose, Object Deformation, H-O Interactions combined, and Camera Motions, all marked with ✓), while other edit types (stylization, attribute modification) are not its primary focus. ✓: Supported. ×: Not Supported. ✓: Partial/Minor Support.

## 2 RELATED WORKS

### 2.1 IMAGE EDITING WITH DIFFUSION

Diffusion models Sohl-Dickstein et al. (2015); Song & Ermon (2019); Ho et al. (2020) have become a cornerstone of text-to-image synthesis, adept at progressively converting random noise into detailed visual content. Building on this foundation, various image editing techniques have emerged. These primarily operate by manipulating sampling trajectories Kim et al. (2022); Meng et al. (2021); Couairon et al. (2022); Mokady et al. (2023); Kwon et al. (2022); Parmar et al. (2023); Huberman-Spiegelglas et al. (2024); Brack et al. (2024), modifying internal model mechanisms or feature representations Hertz et al. (2022); Tumanyan et al. (2023); Cao et al. (2023), or incorporating iterative optimization Kawar et al. (2023); Li et al. (2023); Zhang et al. (2023a). Key strategies include sampling-based methods like SDEdit Meng et al. (2021), which guide image reconstruction from noisy inputs via text prompts, often without retraining. Another prominent category involves attention map manipulation (e.g., Prompt-to-Prompt Hertz et al. (2022), Plug-and-Play Tumanyan et al. (2023), MasaCtrl Cao et al. (2023)). These typically rely on DDIM inversion Song et al. (2020), the quality of which is crucial and has been significantly enhanced by refinement techniques such as Null-text Inversion Mokady et al. (2023), Direct Inversion Ju et al. (2023), ProxEdit Han et al. (2024), and TurboEdit Deutch et al. (2024); Wu et al. (2024).

### 2.2 DATASETS FOR INSTRUCTION-BASED EDITING

Instruction-based image editing models, which aim to efficiently edit real images according to user guidance, rely critically on the quality and scope of their training datasets. Table 1 summarizes popular public available datasets, including InstructPix2Pix Brooks et al. (2023), MagicBrush Zhang et al. (2024), HQ-Edit Hui et al. (2024), EditWorld Yang et al. (2024b), UltraEdit Zhao et al. (2024), and AnyEdit Yu et al. (2024). While some datasets like EditWorld and UltraEdit incorporate human curation to improve quality, many others such as InstructPix2Pix, MagicBrush, and HQ-Edit, rely heavily on automated data-generation pipelines.

These pioneering datasets, however, are limited in the following aspects. **First**, they largely prioritize general image editing tasks (*e.g.*, stylization, attribute modification) which aim to preserve the overall image structure. This prevalent focus means they often neglect critical non-rigid transformations, such as changes in object pose, viewpoint, or dynamic camera movements. This limitation is evident in datasets like SEED-Data-Edit Ge et al. (2024), UltraEdit Zhao et al. (2024), EMU-Edit Sheynin et al. (2024), EditWorld Yang et al. (2024b), and AnyEdit Yu et al. (2024). **Second**, the common practice for generating image pairs involves applying tuning-free editing algorithms to single source images (either real or synthetic). This approach can lead to inconsistent quality in the target images, which may contain artifacts and lack the naturalness inherent in true sequential changes. These collective shortcomings impede the development of models capable of performing robust, realistic, and complex non-rigid edits.

To overcome these deficiencies, ByteMorph-6M introduces a dataset specifically tailored for instruction-based editing involving non-rigid motions. Crucially, image pairs in ByteMorph-6M are

Figure 2: **Overview of Synthetic Data Construction**. Given a source frame extracted from the real video, our pipeline proceeds in three steps. a) A Vision-Language Model (VLM) creates a Motion Caption from the instruction template database to animate the given frame. b) This caption guides a video generation model Seawead et al. (2025) to create a natural transformation. c) We sampled frames uniformly from the generated dynamic videos with a fixed interval and treated each pair of neighbouring frames as an image editing pair. We re-captioned the editing instruction by the same VLM, as well as the general description of each sampled frame (not shown in the figure).

derived in a way that ensures high fidelity and natural consistency, directly addressing the quality concerns of previous datasets.

## 2.3 VIDEO GENERATION FOR IMAGE EDITING

The integration of video data into image editing methodologies remains a relatively nascent area of research. Existing approaches that leverage video can be broadly categorized. One common strategy involves extracting frame pairs from pre-existing videos, often to capture subjects under diverse conditions or to represent changes over time Chen et al. (2024); Alzayer et al. (2024); Shi et al. (2024); Luo et al. (2024). A more targeted example, InstructMove Cao et al. (2024), also extracts frame pairs from real videos but further employs multi-modal Large Language Models (MLLMs) to generate corresponding editing instructions. However, this approach is often hampered by the inaccuracy of these generated instructions, stemming from the inherent difficulties MLLMs face in precisely interpreting complex transformations within dynamic video sequences. An alternative direction reformulates image editing itself as a video generation task. For instance, Frame2Frame Rotstein et al. (2024) generates a video sequence based on a source image and an editing instruction, from which a desired edited frame is then selected. While innovative, this method typically encounters significant challenges in terms of the generation quality of individual frames and overall computational efficiency.

In contrast to these approaches, our work leverages high-fidelity video synthesis as a foundational step for creating a rich data source. We employ Seaweed Seawead et al. (2025), a state-of-the-art and highly efficient video generation model, to synthesize diverse and high-quality videos. These synthesized videos then serve as the primary data for training our baseline model. Our data significantly enhances the editing model's ability to learn and generate realistic camera movements, complex object dynamics, and intricate scene-level transformations. By addressing the data limitations and methodological challenges of prior work, our approach advances the capability of image editing systems to handle dynamic and complex scenarios more effectively.

## 3 DATASET CONSTRUCTION

In this section, we introduce ByteMorph-6M, a novel dataset designed to model image editing as a motion transformation process. Our methodology leverages video generation models to produce natural and coherent transitions between source and target images, ensuring edits are both realistic and motion-consistent. The synthetic data construction for our dataset involves three key stages, as depicted in Figure. 2.

### 3.1 MOTION CAPTION

Conventional text-based image editing methods typically employ a source image $I_{src}$ paired with an editing instruction prompt $P$, which specifies the desired modifications. In contrast, we treat editing as a sequential transformation, introducing a prompt type—Motion Caption ($C_m$)—that explicitly describes the progression of edits over time.

Given a source frame $I_{src}$ extracted from a real video $V_{src}$, we integrate visual features from $I_{src}$ with the editing instruction prompt $P$ from a template prompt library, to formulate $C_m$ which is a narrative description that captures the dynamic evolution of the intended edits. The template prompt library contains diverse motion-aware editing prompts for humans, objects, and camera viewpoints and this strategy was inspired by in-context learning (ICL) techniques Dong et al. (2022), as introduced by Rotstein et al. (2024). This caption generation process is automated by leveraging a Vision-Language Model (VLM), particularly ChatGPT-4o OpenAI (2024) in our case. The VLM generates concise, video-like narratives describing transformations.

## 3.2 VIDEO GENERATION

We utilize Seaweed Seawead et al. (2025), a pre-trained generative video latent diffusion model built upon a transformer architecture, to simulate dynamic editing processes. Seaweed is fine-tuned to generate a coherent video sequence conditioned on both the source image $I_{src}$ and a corresponding image-to-video prompt, which is the motion caption $C_m$ in this case. Initially, $I_{src}$ is encoded into a latent representation. Subsequently, guided by $C_m$, the model performs a denoising operation, synthesizing a temporally coherent frame sequence that embodies the desired editing transformation. The transformer backbone effectively integrates visual and textual inputs, ensuring accurate control and consistency throughout the editing process. Given the video generation model $G$, the video generation can be expressed as: $G(I_{src}, C_m) = V_{gen} = \{F_1, \ldots, F_T\}$, where $V_{gen}$ represents the generated video sequence consisting of $T$ frames, and $F_t$ denotes the frame at timestep $t$.

| | | CLIP-SIM$_{txt}$↑ | CLIP-D$_{txt}$↑ | CLIP-SIM$_{img}$↑ | CLIP-D$_{img}$↑ | VLM-Eval↑ |
|---|---|---|---|---|---|---|
| Camera Zoom | InstructPix2Pix (Brooks et al., 2023) | 0.270 | 0.021 | 0.737 | 0.266 | 42.37 |
| | MagicBrush (Zhang et al., 2024) | 0.311 | 0.002 | 0.907 | 0.202 | 49.37 |
| | UltraEdit (SD3) (Zhao et al., 2024) | 0.299 | 0.000 | 0.864 | 0.249 | 54.74 |
| | AnySD Yu et al. (2024) | 0.309 | 0.001 | 0.911 | 0.182 | 40.92 |
| | InstructMove Cao et al. (2024) | 0.283 | 0.027 | 0.821 | 0.294 | 70.66 |
| | OmniControl Tan et al. (2024) | 0.251 | 0.022 | 0.722 | 0.300 | 45.79 |
| | †InstructMove Cao et al. (2024) | 0.301 | 0.045 | 0.846 | 0.425 | 82.29 |
| | †OmniControl Tan et al. (2024) | 0.310 | 0.039 | 0.801 | 0.414 | 74.15 |
| | †ByteMorpher (Ours) | 0.301 | 0.048 | 0.847 | 0.463 | 84.08 |
| | GT | 0.317 | 0.075 | 0.890 | 1.000 | 87.11 |
| Camera Move | InstructPix2Pix (Brooks et al., 2023) | 0.318 | 0.010 | 0.709 | 0.200 | 32.20 |
| | MagicBrush (Zhang et al., 2024) | 0.317 | 0.009 | 0.913 | 0.195 | 52.63 |
| | UltraEdit (SD3) (Zhao et al., 2024) | 0.306 | 0.012 | 0.885 | 0.240 | 59.01 |
| | AnySD Yu et al. (2024) | 0.318 | 0.010 | 0.909 | 0.200 | 49.37 |
| | InstructMove Cao et al. (2024) | 0.305 | 0.016 | 0.862 | 0.291 | 74.86 |
| | OmniControl Tan et al. (2024) | 0.243 | 0.022 | 0.687 | 0.243 | 16.71 |
| | †InstructMove Cao et al. (2024) | 0.304 | 0.027 | 0.883 | 0.412 | 82.53 |
| | †OmniControl Tan et al. (2024) | 0.298 | 0.025 | 0.891 | 0.304 | 79.26 |
| | †ByteMorpher (Ours) | 0.319 | 0.041 | 0.894 | 0.426 | 84.18 |
| | GT | 0.320 | 0.039 | 0.915 | 1.000 | 86.37 |
| Object Motion | InstructPix2Pix (Brooks et al., 2023) | 0.299 | 0.026 | 0.789 | 0.257 | 36.47 |
| | MagicBrush (Zhang et al., 2024) | 0.328 | 0.007 | 0.901 | 0.163 | 47.49 |
| | UltraEdit (SD3) (Zhao et al., 2024) | 0.324 | 0.012 | 0.887 | 0.237 | 62.13 |
| | AnySD Yu et al. (2024) | 0.319 | 0.008 | 0.879 | 0.189 | 48.31 |
| | InstructMove Cao et al. (2024) | 0.325 | 0.015 | 0.870 | 0.318 | 72.44 |
| | OmniControl Tan et al. (2024) | 0.279 | 0.023 | 0.753 | 0.270 | 34.11 |
| | †InstructMove Cao et al. (2024) | 0.328 | 0.043 | 0.891 | 0.481 | 87.97 |
| | †OmniControl Tan et al. (2024) | 0.330 | 0.036 | 0.892 | 0.470 | 86.48 |
| | †ByteMorpher (Ours) | 0.332 | 0.044 | 0.896 | 0.472 | 89.07 |
| | GT | 0.335 | 0.056 | 0.919 | 1.000 | 89.53 |
| Human Motion | InstructPix2Pix (Brooks et al., 2023) | 0.248 | 0.012 | 0.694 | 0.211 | 23.60 |
| | MagicBrush (Zhang et al., 2024) | 0.317 | 0.001 | 0.911 | 0.146 | 46.27 |
| | UltraEdit (SD3) (Zhao et al., 2024) | 0.313 | 0.011 | 0.900 | 0.195 | 50.64 |
| | AnySD Yu et al. (2024) | 0.312 | 0.003 | 0.894 | 0.156 | 38.12 |
| | InstructMove Cao et al. (2024) | 0.308 | 0.013 | 0.861 | 0.278 | 69.43 |
| | OmniControl Tan et al. (2024) | 0.230 | 0.018 | 0.660 | 0.229 | 25.18 |
| | †InstructMove Cao et al. (2024) | 0.314 | 0.023 | 0.901 | 0.442 | 84.70 |
| | †OmniControl Tan et al. (2024) | 0.311 | 0.016 | 0.880 | 0.399 | 80.78 |
| | †ByteMorpher (Ours) | 0.316 | 0.022 | 0.899 | 0.440 | 85.66 |
| | GT | 0.321 | 0.031 | 0.922 | 1.000 | 86.10 |
| Interaction | InstructPix2Pix (Brooks et al., 2023) | 0.271 | 0.020 | 0.732 | 0.263 | 31.29 |
| | MagicBrush (Zhang et al., 2024) | 0.317 | 0.004 | 0.914 | 0.167 | 39.98 |
| | UltraEdit (SD3) (Zhao et al., 2024) | 0.314 | 0.018 | 0.892 | 0.226 | 52.24 |
| | AnySD Yu et al. (2024) | 0.315 | 0.005 | 0.909 | 0.173 | 37.23 |
| | InstructMove Cao et al. (2024) | 0.309 | 0.019 | 0.855 | 0.318 | 67.07 |
| | OmniControl Tan et al. (2024) | 0.258 | 0.021 | 0.689 | 0.265 | 32.99 |
| | †InstructMove Cao et al. (2024) | 0.314 | 0.043 | 0.885 | 0.477 | 85.83 |
| | †OmniControl Tan et al. (2024) | 0.295 | 0.041 | 0.768 | 0.433 | 78.90 |
| | †ByteMorpher (Ours) | 0.320 | 0.045 | 0.884 | 0.483 | 86.61 |
| | GT | 0.324 | 0.046 | 0.905 | 1.000 | 88.84 |

Table 2: Quantitative Evaluation of open-source methods on ByteMorph-Bench. † represents that the method is trained on ByteMorph-6M. We run each method four times and report the average value. In addition to CLIP similarity metrics, we use Claude-3.7-Sonnet Anthropic (2025) to evaluate the overall editing quality (VLM-Score) on a scale of 0-100. The metrics highlighted in light orange and dark orange are more important.

## 3.3 IMAGE-INSTRUCTION PAIR CREATION

The generated video sequences from Seaweed Seawead et al. (2025) provide a dynamic basis for constructing high-quality image editing pairs. To effectively leverage these videos for training, we uniformly sample four frames $\widetilde{F}_i$ from each synthetic video sequence of length $T = 301$ with fixed intervals of 100 frames, ensuring comprehensive coverage of the editing transformation dynamics. Each pair of sampled neighboring frames $(\widetilde{F}_i, \widetilde{F}_{i+1})$ constitutes a source-target editing pair, thereby forming three distinct editing examples per generated sequence: $(\widetilde{F}_1, \widetilde{F}_2)$, $(\widetilde{F}_2, \widetilde{F}_3)$, and $(\widetilde{F}_3, \widetilde{F}_4)$. These pairs capture incremental but perceptually significant transformations, essential for modeling nuanced non-rigid motions. To avoid the cases where generated videos did not follow the transformation described in the motion caption $C_m$, we utilize the same Vision-Language Model (VLM), ChatGPT-4o OpenAI (2024), to automatically generate precise editing instructions $\widetilde{P}_{i,i+1}$. The VLM examines each frame pair and produces descriptive captions detailing the transformation from the source frame to the target frame. Additionally, a general description of each sampled frame is provided, enriching the dataset with contextual semantic information. This automated captioning ensures consistency, scalability, and semantic clarity across the dataset. Through this method, we effectively construct an extensive, high-quality collection of image-instruction pairs, specifically curated for benchmarking and developing advanced models capable of intricate, motion-guided image editing tasks.

## 3.4 DATASET ANALYSIS

Our dataset contains a total of 6.45 million instruction-based image editing data samples with five motion categories, as shown in Figure. 1. To our knowledge, this is the largest dataset focused on non-rigid image editing to be released to the public (a detailed comparison to other datasets can be found in Table 1). While existing instruction-guided image editing datasets Hui et al. (2024); Zhang et al. (2024); Zhao et al. (2024); Sheynin et al. (2024) primarily emphasize appearance alterations and localized modifications, they rarely contain non-rigid motion transformations. In contrast, ByteMorph-6M is explicitly designed to address this limitation by focusing on dynamic edits such as camera movements, human articulation, and object-object interactions—motions that are underrepresented in current benchmarks. To ensure realism and motion fidelity, ByteMorph-6M comprises high-quality image pairs derived from realistic motion transformations. Furthermore, ByteMorph-6M offers consecutive frames sampled from videos, enabling research into other tasks like multi-reference editing (detailed in Section. 5), expanding beyond the single-reference settings explored in works such as InstructMove Cao et al. (2024) and Frame2Frame Rotstein et al. (2024). Importantly, improving image-based motion editing is not only scientifically valuable but also practically advantageous: image edits execute in seconds, while video generation may take minutes. This advantage of efficiency makes it possible for rapid, flexible customization in real-world applications where user responsiveness and iteration are critical.

## 3.5 MODEL FINETUNING

After construction of ByteMorph-6M, we fine-tuned a Diffusion Transformer Model on the training data with the pre-trained Flux.1-dev Black Forest Labs (2024) text-to-image model as the backbone. We feed the source image and target image into the same VAE encoder and concatenate the noisy source latent and target latent along the sequence length dimension. The position encoding for both latents are shared with exactly the same embeddings. The DiT is fine-tuned with the same loss as the original Flux.1-dev, while all other parameters are frozen. We introduce full training details in Section E of the supplementary materials.

## 4 EXPERIMENTS

### 4.1 IMPLEMENTATION DETAILS

**Settings.** We employ Seaweed Seawead et al. (2025) as the video generation backbone to synthesize image-instruction pairs for ByteMorph-6M. The real videos for extracting source frames were collected from online resources, carefully filtered through video quality assessments and content safety checks. For ByteMorpher, we utilize FLUX.1-dev Black Forest Labs (2024) as the foundational architecture, and train the Diffusion Transformer on 8 NVIDIA H100 GPUs with a total batch size of 8. The training process employs the AdamW optimizer with a learning rate of $1 \times 10^{-5}$. Additionally, we use DeepSpeed Stage 2 to enhance training stability and optimize memory utilization.

**Benchmark Evaluation.** We manually selected 613 high-quality editing pairs, creating the ByteMorph-Bench for a more challenging and comprehensive evaluation. We categorize image editing pairs into five different classes according to the instructions, including 1) camera zoom - the camera position to take the source image zooms in or out, 2) camera move - the camera position to take the source image moves horizontally or vertically, 3) object motion - the object in the source image moves, 4) human motion - the human in the source image moves, and 5) interaction - the objects or humans in the source image interact with each other. **Notably, these pairs are not visible during training.** We also conducted an evaluation on another benchmark with real-world images only, the InstrctMove Cao et al. (2024) benchmark. Both benchmarks evaluate editing models by comparing non-rigid motion edited results with the ground truth.

**Baselines.** For evaluation of existing methods, we adopted the following baselines: 1) Open-source state-of-the-art methods from academia, including InstructPix2Pix Brooks et al. (2023), MagicBrush Zhang et al. (2024), UltraEdit Zhao et al. (2024), AnySD Yu et al. (2024), OminiControl Tan et al. (2024), and InstructMove Cao et al. (2024) 2) Other most recent methods from industry, including Step1X-Edit StepFun (2025), HiDream-E1-Full vivago.ai (2025), Imagen-3-capabilities Google (2025), Gemini-2.0-flash-image-generation Google (2025), SeedEdit 1.6 ByteDance (2024), and GPT-4o-image-generation OpenAI (2025). For those models without publicly available code and weights, we directly call the model APIs following the official instructions.

| | Method | CLIP-SIM$_{txt}$↑ | CLIP-D$_{txt}$↑ | CLIP-SIM$_{img}$↑ | CLIP-D$_{img}$↑ | VLM-Eval↑ | Human-Eval-FL↑ | Human-Eval-ID↑ |
|---|---|---|---|---|---|---|---|---|
| Camera Zoom | Step1X-Edit StepFun (2025) | 0.310 | 0.025 | **0.943** | 0.258 | 59.34 | 26.60 | 48.86 |
| | HiDream-E1-FULL vivago.ai (2025) | 0.304 | 0.027 | 0.682 | 0.287 | 41.18 | 33.00 | 16.50 |
| | Imagen-3-capability Google (2025) | 0.293 | 0.025 | 0.846 | 0.264 | 53.94 | 61.34 | 41.38 |
| | Gemini-2.0-flash-image Google (2025) | 0.305 | 0.031 | 0.862 | 0.297 | 72.27 | 61.04 | 63.09 |
| | SeedEdit 1.6 ByteDance (2024) | 0.311 | 0.029 | 0.827 | 0.325 | 75.00 | 61.34 | **83.60** |
| | GPT-4o-image OpenAI (2025) | **0.317** | 0.015 | 0.832 | 0.337 | 88.14 | **89.36** | 61.09 |
| | BAGEL Deng et al. (2025) | 0.300 | 0.031 | 0.860 | 0.301 | 75.55 | - | - |
| | Flux-Kontext-pro Black Forest Labs (2025) | 0.312 | 0.024 | 0.864 | 0.334 | 75.66 | - | - |
| | Flux-Kontext-max Black Forest Labs (2025) | 0.307 | 0.032 | 0.871 | 0.373 | 80.18 | - | - |
| | SeedEdit 3.0 Wang et al. (2025) | 0.296 | 0.027 | 0.833 | 0.370 | 88.25 | - | - |
| | **ByteMorpher** (Ours) | 0.301 | **0.048** | 0.847 | **0.463** | 84.08 | 61.13 | 74.73 |
| | GT | 0.317 | 0.075 | 0.890 | 1.000 | 87.11 | - | - |
| Camera Move | Step1X-Edit StepFun (2025) | 0.315 | 0.008 | **0.946** | 0.208 | 57.96 | 33.50 | 63.39 |
| | HiDream-E1-FULL vivago.ai (2025) | 0.309 | 0.029 | 0.712 | 0.252 | 32.76 | 16.50 | 18.22 |
| | Imagen-3-capability Google (2025) | 0.282 | 0.010 | 0.813 | 0.238 | 47.22 | 17.38 | 26.51 |
| | Gemini-2.0-flash-image Google (2025) | 0.317 | 0.020 | 0.892 | 0.311 | 77.96 | 56.60 | 75.76 |
| | SeedEdit 1.6 ByteDance (2024) | 0.314 | 0.015 | 0.866 | 0.253 | 78.59 | 58.30 | 87.78 |
| | GPT-4o-image OpenAI (2025) | **0.321** | 0.011 | 0.865 | 0.285 | 84.57 | 76.74 | 59.14 |
| | BAGEL Deng et al. (2025) | 0.306 | 0.026 | 0.883 | 0.290 | 76.08 | - | - |
| | Flux-Kontext-pro Black Forest Labs (2025) | 0.312 | 0.016 | 0.891 | 0.286 | 79.14 | - | - |
| | Flux-Kontext-max Black Forest Labs (2025) | 0.315 | 0.019 | 0.896 | 0.325 | 85.97 | - | - |
| | SeedEdit 3.0 Wang et al. (2025) | 0.308 | 0.020 | 0.887 | 0.278 | 78.00 | - | - |
| | **ByteMorpher** (Ours) | 0.319 | **0.041** | 0.894 | **0.426** | 84.18 | 67.60 | 58.25 |
| | GT | 0.320 | 0.039 | 0.915 | 1.000 | 86.37 | - | - |
| Object Motion | Step1X-Edit StepFun (2025) | 0.323 | 0.019 | **0.923** | 0.260 | 72.78 | 72.16 | 59.39 |
| | HiDream-E1-FULL vivago.ai (2025) | 0.312 | 0.028 | 0.700 | 0.259 | 35.00 | 44.34 | 49.75 |
| | Imagen-3-capability Google (2025) | 0.324 | 0.027 | 0.870 | 0.261 | 57.06 | 62.56 | 77.84 |
| | Gemini-2.0-flash-image Google (2025) | 0.333 | 0.040 | 0.892 | 0.341 | 79.08 | 74.77 | 86.62 |
| | SeedEdit 1.6 ByteDance (2024) | 0.332 | 0.025 | 0.874 | 0.323 | 80.21 | 66.50 | 79.12 |
| | GPT-4o-image OpenAI (2025) | **0.339** | 0.029 | 0.861 | 0.354 | 90.60 | 75.19 | 49.91 |
| | BAGEL Deng et al. (2025) | 0.324 | 0.036 | 0.920 | 0.326 | 74.07 | - | - |
| | Flux-Kontext-pro Black Forest Labs (2025) | 0.321 | 0.018 | 0.893 | 0.314 | 78.41 | - | - |
| | Flux-Kontext-max Black Forest Labs (2025) | 0.325 | 0.025 | 0.888 | 0.353 | 80.42 | - | - |
| | SeedEdit 3.0 Wang et al. (2025) | 0.321 | 0.036 | 0.905 | 0.344 | 88.11 | - | - |
| | **ByteMorpher** (Ours) | 0.332 | **0.044** | 0.896 | **0.472** | 89.07 | 62.16 | 58.25 |
| | GT | 0.335 | 0.056 | 0.919 | 1.000 | 89.53 | - | - |
| Human Motion | Step1X-Edit StepFun (2025) | 0.315 | 0.017 | **0.931** | 0.212 | 65.39 | 44.50 | 78.80 |
| | HiDream-E1-FULL vivago.ai (2025) | 0.301 | 0.017 | 0.676 | 0.215 | 33.21 | 12.51 | 38.66 |
| | Imagen-3-capability Google (2025) | 0.295 | 0.017 | 0.840 | 0.233 | 55.70 | 33.34 | 61.17 |
| | Gemini-2.0-flash-image Google (2025) | 0.314 | 0.017 | 0.893 | 0.282 | 78.72 | 51.84 | 63.34 |
| | SeedEdit 1.6 ByteDance (2024) | 0.324 | 0.024 | 0.878 | 0.274 | 80.62 | 56.23 | 72.12 |
| | GPT-4o-image OpenAI (2025) | 0.316 | 0.021 | 0.850 | 0.330 | 87.93 | **87.56** | 57.84 |
| | BAGEL Deng et al. (2025) | 0.312 | 0.021 | 0.929 | 0.242 | 74.36 | - | - |
| | Flux-Kontext-pro Black Forest Labs (2025) | 0.314 | 0.017 | 0.918 | 0.283 | 79.15 | - | - |
| | Flux-Kontext-max Black Forest Labs (2025) | 0.316 | 0.016 | 0.908 | 0.307 | 80.78 | - | - |
| | SeedEdit 3.0 Wang et al. (2025) | 0.313 | 0.025 | 0.903 | 0.343 | 88.13 | - | - |
| | **ByteMorpher** (Ours) | 0.316 | 0.022 | 0.899 | **0.440** | 85.66 | 68.38 | **75.00** |
| | GT | 0.321 | 0.031 | 0.922 | 1.000 | 86.10 | - | - |
| Interaction | Step1X-Edit StepFun (2025) | 0.312 | 0.020 | **0.937** | 0.245 | 65.99 | 36.09 | 64.56 |
| | HiDream-E1-FULL vivago.ai (2025) | 0.307 | 0.019 | 0.679 | 0.251 | 35.73 | 10.60 | 38.66 |
| | Imagen-3-capability Google (2025) | 0.307 | 0.023 | 0.863 | 0.254 | 54.78 | 47.16 | 61.59 |
| | Gemini-2.0-flash-image Google (2025) | 0.316 | 0.027 | 0.889 | 0.327 | 76.86 | 60.70 | 77.94 |
| | SeedEdit 1.6 ByteDance (2024) | **0.326** | 0.032 | 0.878 | 0.316 | 78.27 | 49.78 | 80.10 |
| | GPT-4o-image OpenAI (2025) | 0.318 | 0.031 | 0.851 | 0.351 | 88.65 | 81.17 | 73.72 |
| | BAGEL Deng et al. (2025) | 0.312 | 0.037 | 0.913 | 0.301 | 73.16 | - | - |
| | Flux-Kontext-pro Black Forest Labs (2025) | 0.313 | 0.028 | 0.898 | 0.318 | 78.58 | - | - |
| | Flux-Kontext-max Black Forest Labs (2025) | 0.320 | 0.032 | 0.894 | 0.335 | 80.12 | - | - |
| | SeedEdit 3.0 Wang et al. (2025) | 0.312 | 0.036 | 0.894 | 0.371 | 86.07 | - | - |
| | **ByteMorpher** (Ours) | 0.320 | **0.045** | 0.884 | **0.483** | 86.61 | 69.15 | 64.73 |
| | GT | 0.324 | 0.046 | 0.905 | 1.000 | 88.84 | - | - |

Table 3: Quantitative Evaluation of ByteMorpher and other methods from the industry on ByteMorph-Bench. We run each method four times and report the average value. In addition to CLIP similarity metrics, we use Claude-3.7-Sonnet Anthropic (2025) to evaluate the overall editing quality (VLM-Score). We also ask human participants to evaluate the instruction-following quality (Human-Eval-FL) and identity-preserving quality (Human-Eval-ID). The final scores from VLM and Human are post-processed and presented on a scale of 0-100 for better understanding. The metrics highlighted in light orange and dark orange are more important.

**Metrics.** Following prior works Zhang et al. (2024); Yu et al. (2024); Cao et al. (2024), we adopt CLIP similarity metrics, e.g., CLIP-SIM$_{img}$, CLIP-SIM$_{txt}$, and CLIP-D$_{txt}$, in our benchmark. Specifically, we denote the source image, ground truth target image, and generated image as $I_{src}$, $I_{tgt}$, and $I_{gen}$, and the text captions of the source and target images as $T_{src}$ and $T_{tgt}$. The details of CLIP-SIM$_{img}$, CLIP-SIM$_{txt}$, and CLIP-D$_{txt}$ are defined in supplementary sections.

However, we observe that these metrics do not always reliably reflect editing quality, particularly for cases involving camera motion. To address this, we introduce a novel yet simple metric, CLIP-D$_{img}$, defined as follow equation with CLIP encoder $\mathbf{C}(\cdot)$:

$$\text{CLIP-D}_{\text{img}} = \cos\left(\mathbf{C}(I_{\text{gen}}) - \mathbf{C}(I_{\text{src}}),\ \mathbf{C}(I_{\text{tgt}}) - \mathbf{C}(I_{\text{src}})\right) \qquad \text{(Eq. 1)}$$

Given that the image editing pairs in our benchmark are manually curated and quality-checked, this metric offers a more accurate measure of overall editing quality. For evaluation on the Instruct-Move Cao et al. (2024) benchmark, we directly use the original test set and metrics. To ensure the validity of our experimental results, we evaluate all methods by running the generation process **four** times on the benchmarks and calculating the average value of the resulting metrics.

**VLM Evaluation and Human Preference Study.** For VLM evaluation, we used the Claude-3.7-Sonnet Anthropic (2025), instead of the GPT OpenAI (2024), or Gemini Team et al. (2023) series for

fair comparison and to avoid possible leakage issues. For human evaluation, we asked 40 participants to judge the Instruction-Following quality (Human-Eval) and ID-Preserving quality (Human-Eval-ID). The details are specified in Section. B of the supplementary materials.

## 4.2 EVALUATION

**Quantitative Results.** We conduct comprehensive evaluations of both open-source methods (Table 2) and industrial-grade models (Table 3) on ByteMorph-Bench. For interpretability, both VLM and human evaluation scores are normalized to a 0–100 scale. ByteMorpher achieves state-of-the-art performance among open-source methods, demonstrating strong instruction-following capabilities and consistent identity preservation across all editing categories. Compared to commercial models, it also delivers competitive overall performance and exhibits a more favorable balance between instruction fidelity and identity retention. These results underscore the effectiveness of our proposed ByteMorph-6M and highlight its value as a rigorous benchmark for evaluating non-rigid motion editing. Among industrial methods, GPT-4o-image OpenAI (2025) excels in instruction-following, as reflected by its high Human-Eval-FL scores. However, it underperforms in identity preservation compared to SeedEdit-1.6 ByteDance (2024) and Gemini-2.0-flash-image Google (2025), both of which maintain stronger visual consistency with the source content. HiDream-E1-Full vivago.ai (2025) and Step1X-Edit StepFun (2025), which are partially open-sourced with inference-only access, show relatively weak performance. We hypothesize that this is due to a lack of training data with diverse and fine-grained motion transformations, such as those found in ByteMorph-6M.

**Qualitative Results.** We present qualitative comparisons on the ByteMorph-Bench in Figure 3. Both GPT-4o-Image OpenAI (2025) and ByteMorpher exhibit strong instruction-following capabilities; however, GPT-4o-Image often struggles to maintain the identity and appearance of subjects in the source image. In contrast, Gemini-2.0-flash-image Google (2025) and SeedEdit-1.6 ByteDance (2024) are more ef-

| | CLIP-$D_{txt}$↑ | CLIP-$SIM_{txt}$↑ | CLIP-$SIM_{img}$↑ |
|---|---|---|---|
| NullTextInversion Mokady et al. (2023) | 0.0660 | 0.7648 | 0.9063 |
| MasaCtrl Cao et al. (2023) | 0.0436 | 0.8527 | 0.9160 |
| InstructPix2Pix Brooks et al. (2023) | 0.0887 | 0.8569 | **0.9380** |
| UltraEdit Zhao et al. (2024) | 0.0824 | 0.8571 | 0.9184 |
| MagicBrush Zhang et al. (2024) | 0.0972 | 0.8648 | _0.9318_ |
| InstrcutMove Cao et al. (2024) | **0.1361** | _0.8724_ | 0.9275 |
| OmniControl Tan et al. (2024) | 0.1089 | 0.8359 | 0.7909 |
| AnySD Yu et al. (2024) | 0.0463 | 0.8415 | 0.9240 |
| **ByteMorpher** (Ours) | _0.1335_ | **0.8784** | 0.9083 |

Table 4: Quantitative comparison with state-of-the-art text-guided image editing methods on Instruct-Move non-rigid image editing benchmark.

fective at preserving identity, but frequently fail to accurately execute the motion-editing instructions. Additional qualitative results are provided in the supplementary materials.

**Ablation Study.** To validate the effectiveness of ByteMorph-6M, we fine-tune OminiControl Tan et al. (2024) and InstructMove Cao et al. (2024) on our training set and report the performance in Table 4. Both models exhibit notable gains across key metrics after fine-tuning. Qualitative results are provided in Figure 4, demonstrating that models trained on ByteMorph-6M achieve substantially better instruction-following ability, particularly for non-rigid motion edits.

## 4.3 LIMITATIONS

Since the training and evaluation data in ByteMorph-6M are collected from both generated and real-world videos featuring motion changes, variations in object stylization, and relighting conditions are limited. Expanding the dataset with additional data from simulators or rendering engines would significantly increase its diversity and scale.

## 5 POTENTIAL USAGE AND FUTURE

**Multi-Reference Editing and Image Reasoning.** As visualized in Figure 1, we sampled multiple images in a single video and performed the instruction labeling separately for each pair of images. It is possible to use our training set for multi-reference image editing, which is an underexplored domain. It's also possible to create models for consecutive motion editing with our dataset, with a reasoning model conditioned on previous editing instructions, source images, and corresponding image captions. We leave these tasks to the community for further exploration. We also discuss more about the potential evaluation metrics on Image Reasoning in Section. C of the supplementary materials.

**Addressing Key Challenges.** As mentioned in Section 4, existing methods face significant limitations, particularly in identity preservation and instruction adherence. To explore viable solutions for improving identity-aware editing, we propose two strategies: (1) *Identity Embedding Consistency*. Incorporate precomputed identity embeddings by leveraging frozen encoders—such as ArcFace

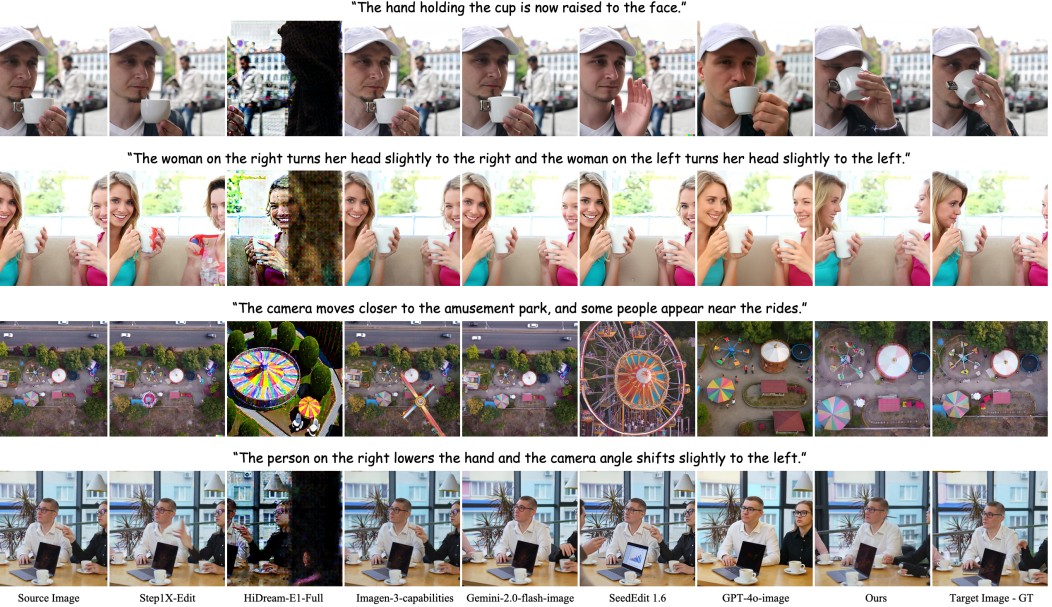

Source Image · Step1X-Edit · HiDream-E1-Full · Imagen-3-capabilities · Gemini-2.0-flash-image · SeedEdit 1.6 · GPT-4o-image · Ours · Target Image - GT

Figure 3: **Qualitative comparison of ByteMorpher and other methods from the industry on ByteMorph-Bench.** GPT-4o-Image and ByteMorpher demonstrate strong instruction-following quality while GPT-4o-Image struggles to preserve the identity and appearance in the source image. On the other hand, Gemini-2.0-flash-image and SeedEdit 1.6 preserve ID well but cannot consistently follow the motion editing instructions.

for human faces or DINO for general objects—and apply a consistency loss between the source and generated images during fine-tuning. This encourages the model to retain critical identity features. (2) *Multi-Stage Refinement*. Generate a coarse motion-editing output in the first stage, followed by identity-preserving refinement through a second-stage module, such as an image fusion network or denoising model. This process can be enhanced using region-specific attention or cross-attention mechanisms aligned with the original image features. To improve instruction-following, especially for non-rigid camera motions, we explore two complementary approaches: (1) *Text-to-Motion Embedding with Structural Guidance*. Develop a motion grounding module that maps natural language instructions to spatial motion vectors (e.g., using the Plücker embedding Sitzmann et al. (2021)), which can be integrated into the diffusion backbone. This provides explicit motion cues, similar to strategies in MotionCtrl Wang et al. (2024), CameraCtrl He et al. (2024), and CameraCtrl II He et al. (2025). (2) *Instruction Perturbation Training*. Augment the training set with semantically equivalent instruction variants (e.g., "move camera to the left" vs. "shift view leftward") to improve robustness and generalization in instruction understanding.

## 6 CONCLUSION

In this work, we introduce ByteMorph, a new framework for expressive image editing with non-rigid motions, bridging the gap between traditional instruction-based editing and dynamic, motion-centric transformations. To this end, we propose ByteMorph-6M, a large-scale dataset designed to capture a wide spectrum of non-rigid editing scenarios, including camera movements, object deformations, human articulations, and human-object interactions. Extensive experiments on ByteMorph-Bench demonstrate that new challenges that are not addressed in previous datasets exist.

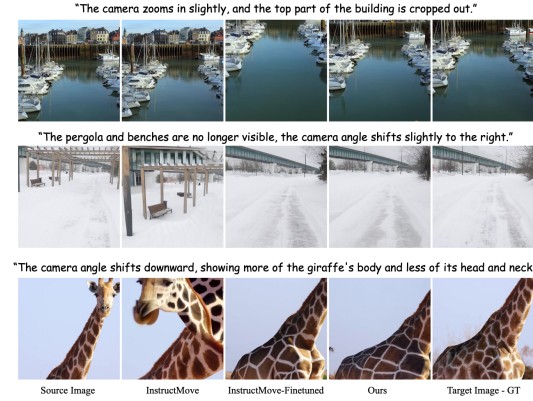

Source Image · InstructMove · InstructMove-Finetuned · Ours · Target Image - GT

Figure 4: **Visualization of ablation analysis.** InstrucMove-Finetuned denotes we fine-tuned InstructMove Cao et al. (2024) on the training set of ByteMorph-6M. After such fine-tuning, the instruction-following quality for camera motion editing is significantly improved.

**Ethics statement.** Our work aims to improve instruction-based image editing from a technical perspective and is not intended for misuse, such as forgery. Therefore, synthesized images should clearly indicate their artificial nature. This work is for research purposes only, and the usage of our dataset should strictly follow the rules specified in the license and guidelines.

**Reproducibility statement.** The proposed benchmark ByteMorph-Bench, dataset ByteMorph-6M, model ByteMorpher will be open-source upon acceptance. We also released them with full instructions in anonymous downloadable sources as detailed in Section. A of the appendix.

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

## A  RELEASE OF DATASET AND BENCHMARK CODE

We have released the full training set, evaluation benchmark, and baseline code. They are published in the following anonymous links:

- Train Data:
  `https://huggingface.co/datasets/ByteMorph/BM-6M`

- Evaluation Benchmark:
  `https://huggingface.co/datasets/ByteMorph/BM-Bench`

- Evaluation Code and Instructions:
  `https://github.com/ByteMorph/BM-code/tree/main/`
  `ByteMorph-Eval`

- Baseline Model Train and Inference Code:
  `https://github.com/ByteMorph/BM-code`

- Baseline Model Weight:
  `https://huggingface.co/ByteMorph/BM-Model`

- Project Page:
  `https://bytemorph9.github.io/`

- Leaderboard:
  `https://bytemorph9.github.io/#leaderboard`

ByteMorph-6M and ByteMorph-Bench are released under CC0-1.0 Creative Commons Zero v1.0 Universal License. The baseline model ByteMorpher, including code and weights, is released under FLUX.1-dev Non-Commercial License.

## B  DETAILS OF HUMAN PREFERENCE STUDY

We use **Prolific**, an online platform designed to connect academic researchers with user research participants for human-level performance evaluation. The participants are English-speaking lay people verified by this platform without prior knowledge of computer vision, and they are paid 15 USD/hr for their labor. We randomly chose ten examples from each editing category in our benchmark (50 examples in total). The results generated from different methods are anonymized and randomly sorted in each example. We then asked 40 candidates to choose and rank the **top three best** image editing results from the anonymized methods, for Instruction-Following quality and ID-Preserving ability. We collect their answers and rescale the results back to 0-100, *e.g.*, the First choice = 100, the Second choice = 67, the third choice = 33, and others = 0. The participant may leave the choices blank if there are not enough methods that provide satisfying results, and the participant may choose multiple methods with the same ranking (still up to three in total). A screenshot of our human preference study is presented in Figure. 5.

## C  POTENTIAL FUTURE RESEARCH

As mentioned in Section. 5, we expect the community to utilize our work for further research on multi-frame consecutive editing/reasoning, or multi-reference editing. To evaluate the consistency of multi-frame editing for further research, we believe it's possible to collect consecutive frames and editing prompts (same as collecting training data) as evaluation data, and we can evaluate the consistency by,

1. calculating the editing-step-wise transition consistency

$$TransitionError = |LPIPS(G_i, G_{i-1}) - LPIPS(T_i, T_{i-1})| \qquad (1)$$

   Where $G_i$ is the $i^{th}$ image in the Generated image sequence and $T_i$ is the $i^{th}$ image in the Target ground truth image sequence. In this way, a low average TransitionError across all steps indicates that the model is applying edits with a similar perceptual magnitude as the ground truth.

Please **choose and rank** the **three** best generated Target Image from A to G, given the Source Image on the left and the Edit Instruction.
According to the following criteria:

1. Instruction Following: A successful editing should **not** miss any change required by Edit Instruction, and **not** have any extra changes that are not required by Edit Instruction.
2. ID Preserving: The object or human in the Target Image should maintain the **Same Appearance and Identity** as the Source Image.

**Edit Instruction: The camera zooms in on the person, making them appear closer and more detailed.**

| | A | B | C | D | E | F | G |
|---|---|---|---|---|---|---|---|
| Instruction Following - Best | ◯ | ◯ | ◯ | ◯ | ◯ | ◯ | ◯ |
| Instruction Following - 2nd Best | ◯ | ◯ | ◯ | ◯ | ◯ | ◯ | ◯ |
| Instruction Following - 3rd Best | ◯ | ◯ | ◯ | ◯ | ◯ | ◯ | ◯ |
| ID Preserving - Best | ◯ | ◯ | ◯ | ◯ | ◯ | ◯ | ◯ |
| ID Preserving - 2nd Best | ◯ | ◯ | ◯ | ◯ | ◯ | ◯ | ◯ |
| ID Preserving - 3rd Best | ◯ | ◯ | ◯ | ◯ | ◯ | ◯ | ◯ |

Figure 5: **A screenshot of one example in our human preference study.** The user may leave the choices blank if there are not enough methods that provide satisfying results, and the user may choose multiple methods with the same ranking.

| | Message |
|---|---|
| **System Prompt** | You are an annotator for image editing. You will be given an input image. You need to imagine a similar image editing prompt as the example image editing operation, which is suitable for the given image. The imagined editing prompt should describe at least one of these five categories: 
 1) The human in the given image has motion changes. 2) The object in the given image has motion changes. 3) The camera position used to take the given image zooms in or zooms out. 4) The camera position used to take the given image moves to the left or right or up or down. 5) The object or the human in the given image interacts with each other. The editing prompt should describe one of the five above categories. Here is the example editing operation: {example} 
 Provide your response in a JSON format as such: {"motion_caption": "xxx"} 
 Do not output anything else. |
| **Human Prompt** | {base64_source_image_string} + {example} |
| **Output Example** | {"motion_caption": "The camera angle shifts to the right and the girl on the left opens her eyes."} |

Table 5: An example of Motion Caption. The system prompt describes the VLM's responsibility as an annotator for image editing and defines the Motion Caption task. The human prompt provides the actual input to the VLM from the user. The source image is encoded as a 64-byte string according to GPT-4o OpenAI (2024) API instructions. "example" denotes a randomly selected prompt from the template library. The output caption is saved as a JSON dictionary.

2. identity preservation over editing-steps with embeddings from a pre-trained vision model like DINOv2 (for object and scene) or a face recognition model like ArcFace (for human) to calculate the cosine similarity between generated frames. A higher similarity indicates that the model preserves identity better over consecutive editing steps.

# D  DETAILS ON MOTION CAPTION AND VIDEO GENERATION

For preprocessing of the intial frames input to the video generation model, which are sampled from real videos, we filter the motion blur and artifacts by using the quality assessment metrics e.g. "liqe" "raft" "clipimage" filters to carefully filter out the input reference frames that didn't meet the satisfying image quality. Specifically,1) "liqe" means the selected frames from the real video should have good perceptual quality without blurring or other artifacts 2) "raft" filter calculates the normalization of optical flow between selected frames and filters out image pairs that have very little motion. 3) "clipimage" means the selected frame should align with the original video caption. We then feed these reference frames each by each into the video generation model together with the motion prompt. For the "artifacts are described in the final re-captioned image editing prompts", we believe it's a good thing to keep it in the dataset since they can be used as negative prompts to finetune the ByteMorpher or other user models, so that the final finetuned model can avoid generate such artifacts by learning from our dataset, except explicitly defined by editing prompt.

In Table. 5, we provide an example for Motion Caption mentioned in Section. 3.1.

In Table. 6, we provide an example for Image-Instruction Pair Creation, mentioned in Section. 3.3.

# E  DETAILS ON BYTEMORPHER TRAINING

The overall fine-tuning pipeline of ByteMorpher is shown in Figure 6.

# F  DETAILS ON BYTEMORPH-BENCH EVALUATION METRICS

**CLIP metrics.** CLIP-SIM$_{img}$ measures the cosine similarity between CLIP embedding of source image and generated image as shown in Eq. 1, CLIP-SIM$_{txt}$ evaluates the similarity between

| | Message |
|---|---|
| **System Prompt - 1** | You are given a pair of video frames. You need to observe the differences between the two frames. Then, summarize the changes happened between the frames in one sentence. Do not use past tense. Try to mention all the changes, including the motions of human body or object, the facial expressions of human, and those changes caused by camera movement.
If there are objects and humans interact with each other, explicitly describe the interaction as well.
Only reply the changes. Do not reply anything else. |
| **Human Prompt - 1** | {base64_source_image_string} + {base64_target_image_string} |
| **Output Example - 1** | {"edit_prompt": "The camera zooms out, several people appear on the poolside area, and a potted plant is added near the pool." } |
| **System Prompt - 2** | You are given an image. You need to describe the content in this image. Do not use the past tense. Try to mention all the subjects in this frame, including human or object. Do not output anything else. Directly describe the image, Do not use "This image ..." or "This frame ..." |
| **Human Prompt - 2** | {base64_source_image_string} |
| **Output Example - 2** | {"src_img_caption": "A large outdoor swimming pool with clear blue water is surrounded by a paved deck area. Multiple black lounge chairs are arranged in rows along the sides of the pool. A small building with a door and a potted plant is visible in the corner. A metal pool ladder is positioned at the edge of the pool. In the background, a cityscape with buildings and towers is visible under a partly cloudy sky." } |
| **System Prompt - 3** | You are given an image. You need to describe the content in this image. Do not use the past tense. Try to mention all the subjects in this frame, including human or object. Do not output anything else. Directly describe the image, Do not use "This image ..." or "This frame ..." |
| **Human Prompt - 3** | {base64_target_image_string} |
| **Output Example - 3** | {"tgt_img_caption": "A large outdoor swimming pool is surrounded by a paved area with lounge chairs neatly arranged along one side. Several women are engaged in a group exercise session on the paved area near the pool, wearing athletic clothing in various colors such as red, pink, black, and neon green. A small potted plant is placed near the pool, and the background includes a view of distant buildings and a clear sky with scattered clouds." } |

Table 6: An example of Image-Instruction Pair Creation with **three** separate steps (denoted as **- 1**, **- 2**, and **- 3**. In the first step, the system prompt describes the VLM's responsibility as an annotator for observing the difference between source and target images, and summarizing the editing instructions. The human prompt provides the actual input to the VLM from the user. Both source and target images are encoded as 64-byte strings according to GPT-4o OpenAI (2024) API instructions. The output editing instruction is saved as a JSON dictionary.
In the second and third steps, the system prompt describes the VLM's responsibility as an annotator for describing the source or target image content. The human prompt provides the 64-byte encoded image string to the VLM from the user. The output caption is saved as a JSON dictionary.

generated image and text caption of target image (Eq. 2), and CLIP-D$_{txt}$ captures the directional alignment between CLIP distance between text embedding from source to target and image from source to generation, as defined in Eq. 3. Our proposed metric CLIP-D$_{img}$ is defined in Eq. 4.

$$\text{CLIP-SIM}_{\text{img}} = \cos\left(\mathbf{C}(I_{\text{gen}}),\ \mathbf{C}(I_{\text{src}})\right) \tag{Eq. 1}$$

$$\text{CLIP-SIM}_{\text{txt}} = \cos\left(\mathbf{C}(I_{\text{gen}}),\ \mathbf{C}(T_{\text{tgt}})\right) \tag{Eq. 2}$$

$$\text{CLIP-D}_{\text{txt}} = \cos\left(\mathbf{C}(I_{\text{gen}}) - \mathbf{C}(I_{\text{src}}),\ \mathbf{C}(T_{\text{tgt}}) - \mathbf{C}(T_{\text{src}})\right) \tag{Eq. 3}$$

$$\text{CLIP-D}_{\text{img}} = \cos\left(\mathbf{C}(I_{\text{gen}}) - \mathbf{C}(I_{\text{src}}),\ \mathbf{C}(I_{\text{tgt}}) - \mathbf{C}(I_{\text{src}})\right) \tag{Eq. 4}$$

where $\mathbf{C}(\cdot)$ stands for CLIP encoder.

**VLM-Score.** In Table. 7, we provide the detailed steps for VLM-Score. evaluation with Claude-3.7-Sonne Anthropic (2025).

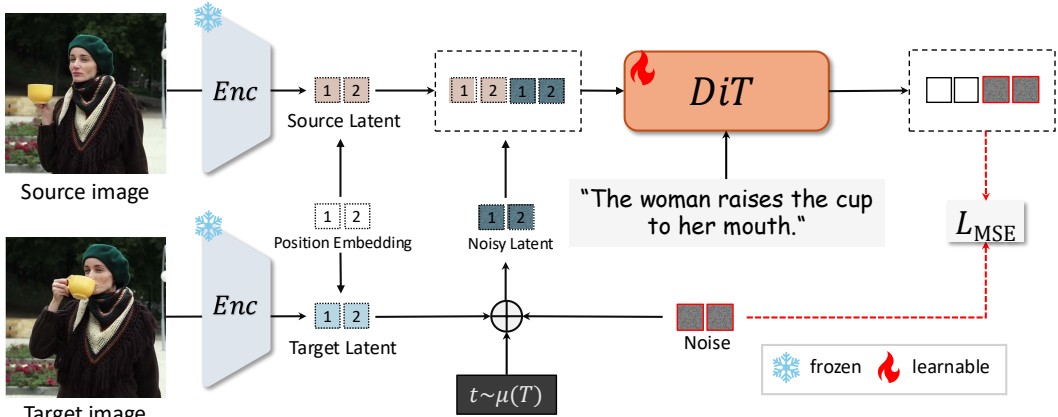

Figure 6: **Overview of ByteMorpher's training.** We fine-tuned the Diffusion Transformer backbone from the pre-trained Flux.1-dev (Black Forest Labs, 2024) text-to-image model on the ByteMorph-6M. We feed the source image and target image into the same frozen VAE encoder and obtain source and target latents. The position encoding for both latents is shared with exactly the same embeddings. The target latent is added with noise and further concatenated with the source latent along the sequence length dimension. The DiT is fine-tuned with the MSE loss, where the difference between the noise and the latter half of the output from DiT (corresponding to the noisy target latent input) is minimized.

| | Message |
|---|---|
| **System Prompt** | You are an evaluator for image editing. You will be given a pair of images before and after editing as well as an editing instruction. You need to rate the editing result with a score between 0 to 100. A successful editing should not miss any change required by editing instruction. A successful editing should not have any extra changes that are not required by editing instruction. The second image should have minimum change to reflect the changes made with editing instruction. Be strict about the changes made between two images. Give the final response in a json format as such: {"Score": xx} Do not output anything else. |
| **Human Prompt** | {base64_source_image_string} + {base64_target_image_string} + {editing_instruction} |
| **Output Example** | {"Score": 90} |

Table 7: An example of VLM-Score Evaluation. The system prompt describes the VLM's responsibility as an evaluator and defines the task. The human prompt provides the actual input to the VLM from the user. Both source and target images are encoded as 64-byte strings according to Claude-3.7-Sonnet Anthropic (2025) API instructions. The output score is saved as a JSON dictionary.

## G   MORE EVALUATION ON BYTEMORPH-BENCH

In addition to the metrics used in Section. , e.g. CLIP scores, VLM judgments, and human preference studies, we also calculate PSNR, SSIM, MSE as geometric metrics to evaluate pixel-aligned error between the GT Image and generated image from different methods. The results is shown in Table. 8 and Table. 9:

## H   THE USE OF LARGE LANGUAGE MODELS (LLMS)

In addition to the usage of LLMs in our dataset construction pipeline, we have used LLMs to aid or polish writing of the paper. We hereby confirm this according to the ICLR Author Guide.

| | Method | PSNR ↑ | SSIM ↑ | MSE ↓ |
|---|---|---|---|---|
| **Camera Zoom** | InstructPix2Pix | 11.6707 | 0.3849 | 0.0929 |
| | MagicBrush | 13.3401 | 0.3924 | 0.0607 |
| | UltraEdit (SD3) | 13.0231 | 0.3970 | 0.0652 |
| | AnySD | 13.2615 | 0.3939 | 0.0620 |
| | InstructMove | 10.9644 | 0.3288 | 0.0987 |
| | OmniControl | 10.2260 | 0.3492 | 0.1093 |
| | **ByteMorpher (Ours)** | **14.6611** | **0.4387** | **0.0443** |
| **Camera Motion** | InstructPix2Pix | 11.9410 | 0.3412 | 0.0728 |
| | MagicBrush | 12.7441 | 0.3373 | 0.0612 |
| | UltraEdit (SD3) | 12.6164 | 0.3358 | 0.0639 |
| | AnySD | 12.4282 | 0.3290 | 0.0677 |
| | InstructMove | 11.2232 | 0.2864 | 0.0852 |
| | OmniControl | 9.6806 | 0.3191 | 0.1180 |
| | **ByteMorpher (Ours)** | **14.2361** | **0.3783** | **0.0433** |
| **Object Motion** | InstructPix2Pix | 14.2845 | 0.4585 | 0.0502 |
| | MagicBrush | 16.0478 | 0.4922 | 0.0377 |
| | UltraEdit (SD3) | 15.5830 | 0.4878 | 0.0388 |
| | AnySD | 15.7905 | 0.4748 | 0.0481 |
| | InstructMove | 13.1264 | 0.3818 | 0.0664 |
| | OmniControl | 10.4842 | 0.3527 | 0.1036 |
| | **ByteMorpher (Ours)** | **17.7158** | **0.5532** | **0.0266** |
| **Human Motion** | InstructPix2Pix | 12.9218 | 0.4214 | 0.0777 |
| | MagicBrush | 15.3138 | 0.4877 | 0.0441 |
| | UltraEdit (SD3) | 15.3151 | 0.5023 | 0.0360 |
| | AnySD | 14.7906 | 0.4730 | 0.0518 |
| | InstructMove | 12.4081 | 0.3952 | 0.0709 |
| | OmniControl | 9.2151 | 0.3917 | 0.1377 |
| | **ByteMorpher (Ours)** | **16.6347** | **0.5372** | **0.0268** |
| **Interaction** | InstructPix2Pix | 12.0986 | 0.4056 | 0.0780 |
| | MagicBrush | 13.9085 | 0.4501 | 0.0553 |
| | UltraEdit (SD3) | 13.7349 | 0.4507 | 0.0501 |
| | AnySD | 13.7381 | 0.4503 | 0.0590 |
| | InstructMove | 11.4411 | 0.3730 | 0.0850 |
| | OmniControl | 9.4863 | 0.3730 | 0.1240 |
| | **ByteMorpher (Ours)** | **15.5847** | **0.5101** | **0.0342** |

Table 8: Quantitative comparison between open-source methods.

## I ADDITIONAL VISUALIZATION

We provide additional visualized comparison between commercial methods StepFun (2025); vivago.ai (2025); Google (2025); ByteDance (2024); Google (2025); OpenAI (2025) and our proposed ByteMorpher in Figure. 7, Figure. 8, Figure. 9, Figure. 10, and Figure. 11. GPT-4o-image and ByteMorpher both demonstrated strong instruction-following ability, while GPT-4o-image performed worse than the Gemini-2.0-flash-image, SeedEdit 1.6, and ByteMorpher in identity preservation.

| | Method | PSNR ↑ | SSIM ↑ | MSE ↓ |
|---|---|---|---|---|
| Camera Zoom | Step1X-Edit | 13.2516 | 0.3985 | 0.0686 |
| | HIDream-E1-FULL | 7.7711 | 0.1460 | 0.1779 |
| | Imagen-3-capability | 12.3923 | 0.3616 | 0.0814 |
| | Gemini-2.0-flash-image | 11.5916 | 0.3380 | 0.0878 |
| | SeedEdit 1.6 | 12.6316 | 0.3525 | 0.0679 |
| | GPT-4o-image | 11.9522 | 0.3349 | 0.0750 |
| | **ByteMorpher (Ours)** | **14.6611** | **0.4387** | **0.0443** |
| Camera Motion | Step1X-Edit | 12.8042 | 0.3424 | 0.0610 |
| | HIDream-E1-FULL | 7.9841 | 0.1343 | 0.1711 |
| | Imagen-3-capability | 10.8464 | 0.2592 | 0.1026 |
| | Gemini-2.0-flash-image | 11.8650 | 0.3110 | 0.0778 |
| | SeedEdit 1.6 | 12.3289 | 0.3115 | 0.0665 |
| | GPT-4o-image | 12.0259 | 0.3160 | 0.0684 |
| | **ByteMorpher (Ours)** | **14.2361** | **0.3783** | **0.0433** |
| Object Motion | Step1X-Edit | 15.8718 | 0.5208 | 0.0374 |
| | HIDream-E1-FULL | 8.1488 | 0.1596 | 0.1666 |
| | Imagen-3-capability | 14.2607 | 0.4586 | 0.0575 |
| | Gemini-2.0-flash-image | 15.4684 | 0.4856 | 0.0417 |
| | SeedEdit 1.6 | 15.3076 | 0.4404 | 0.0397 |
| | GPT-4o-image | 14.1279 | 0.3973 | 0.0475 |
| | **ByteMorpher (Ours)** | **17.7158** | **0.5532** | **0.0266** |
| Human Motion | Step1X-Edit | 15.2873 | 0.5127 | 0.0365 |
| | HIDream-E1-FULL | 7.9548 | 0.1664 | 0.1736 |
| | Imagen-3-capability | 13.3201 | 0.4271 | 0.0643 |
| | Gemini-2.0-flash-image | 15.1715 | 0.4856 | 0.0384 |
| | SeedEdit 1.6 | 14.8531 | 0.4531 | 0.0384 |
| | GPT-4o-image | 13.8688 | 0.4222 | 0.0468 |
| | **ByteMorpher (Ours)** | **16.6347** | **0.5372** | **0.0268** |
| Interaction | Step1X-Edit | 13.9983 | 0.4727 | 0.0483 |
| | HIDream-E1-FULL | 7.5533 | 0.1625 | 0.1886 |
| | Imagen-3-capability | 12.6187 | 0.4026 | 0.0669 |
| | Gemini-2.0-flash-image | 13.7287 | 0.4481 | 0.0505 |
| | SeedEdit 1.6 | 13.8038 | 0.4264 | 0.0490 |
| | GPT-4o-image | 12.9125 | 0.4046 | 0.0573 |
| | **ByteMorpher (Ours)** | **15.5847** | **0.5101** | **0.0342** |

Table 9: Quantitative comparison between methods from the industry.

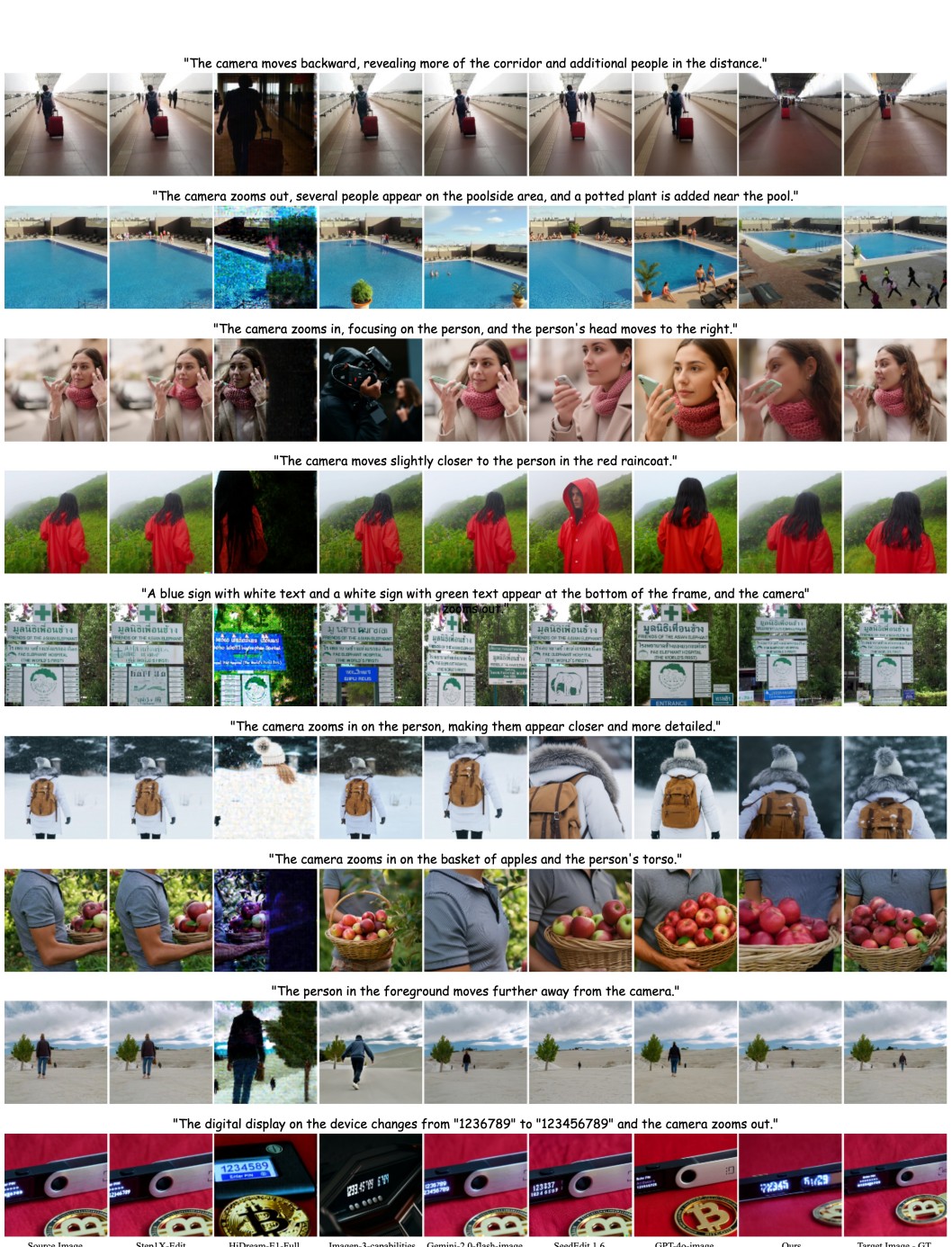

Figure 7: **Visualized comparison between methods from the industry on ByteMorph-Bench.** Editing Category: Camera Zoom.

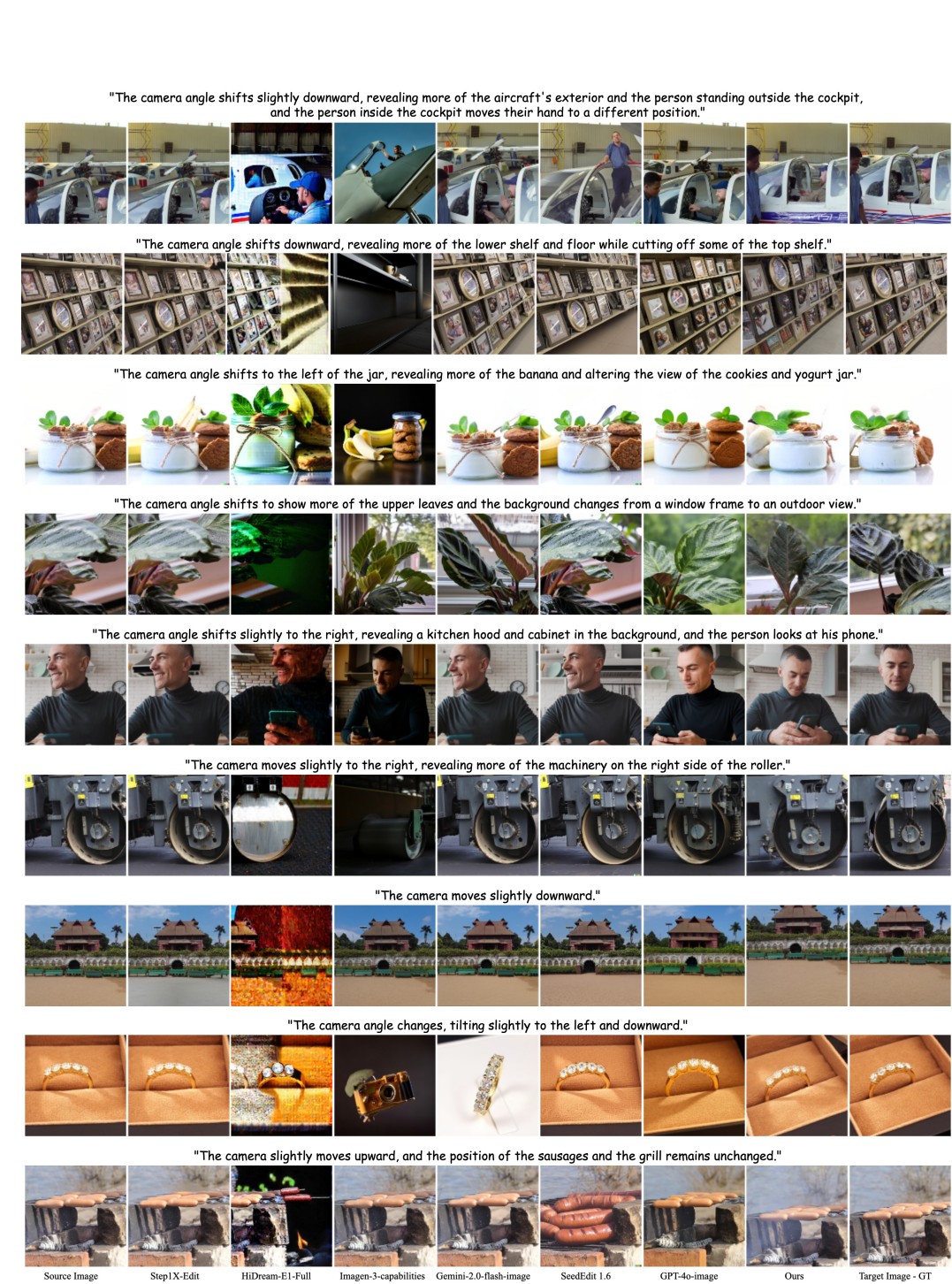

Figure 8: **Visualized comparison between methods from the industry on ByteMorph-Bench.** Editing Category: Camera Motion.

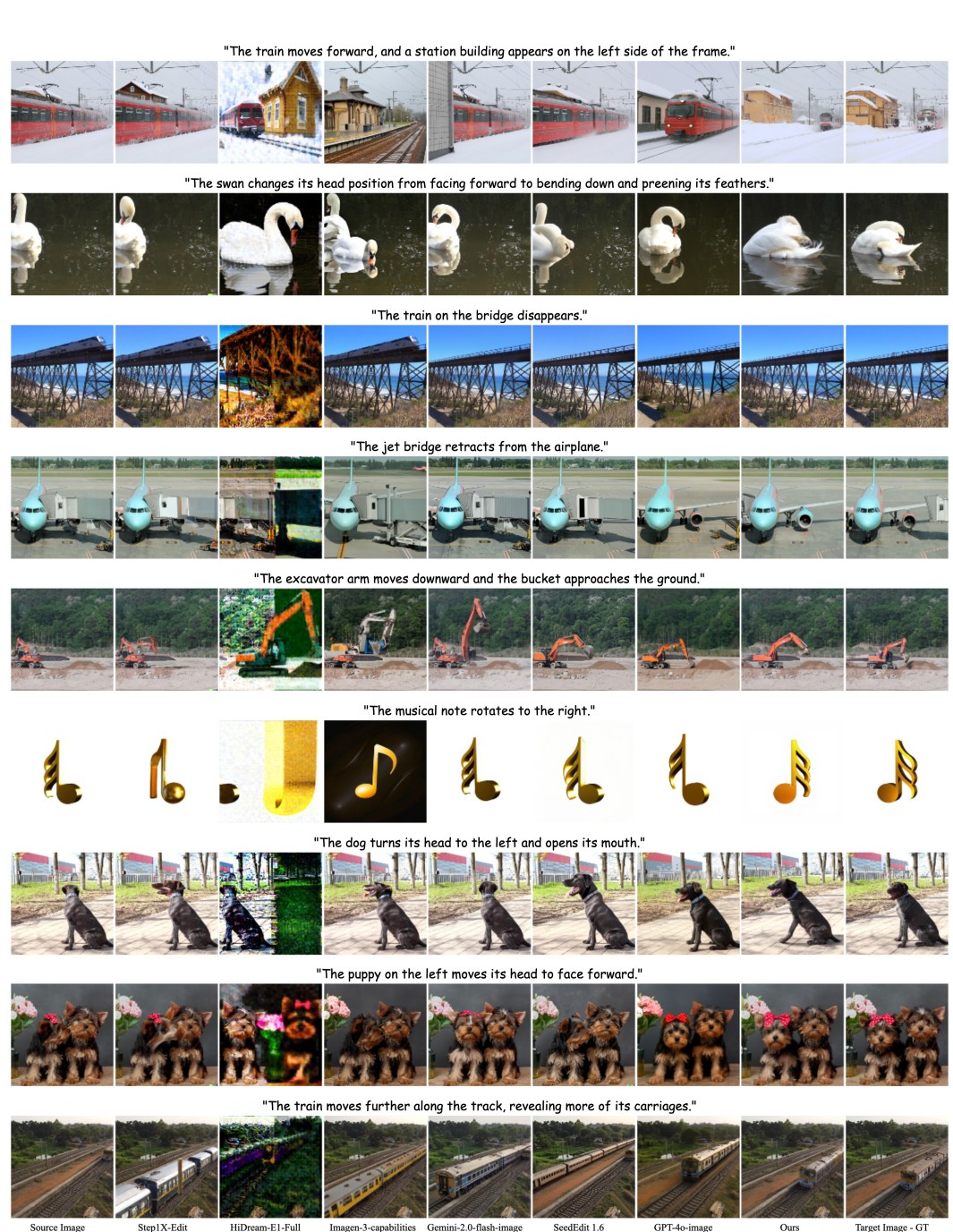

Figure 9: **Visualized comparison between methods from the industry on ByteMorph-Bench.** Editing Category: Object Motion.

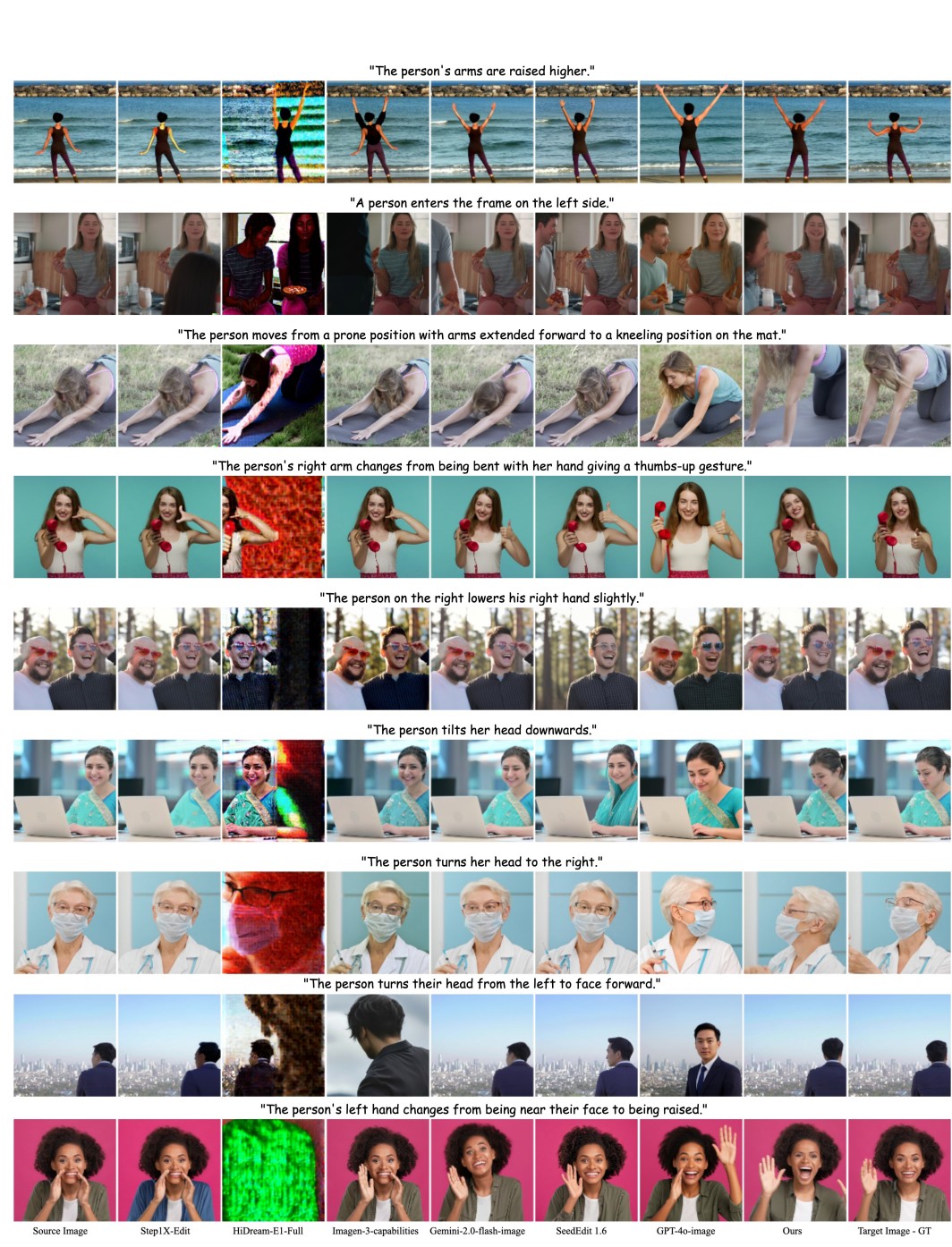

Figure 10: **Visualized comparison between methods from the industry on ByteMorph-Bench.** Editing Category: Human Motion.

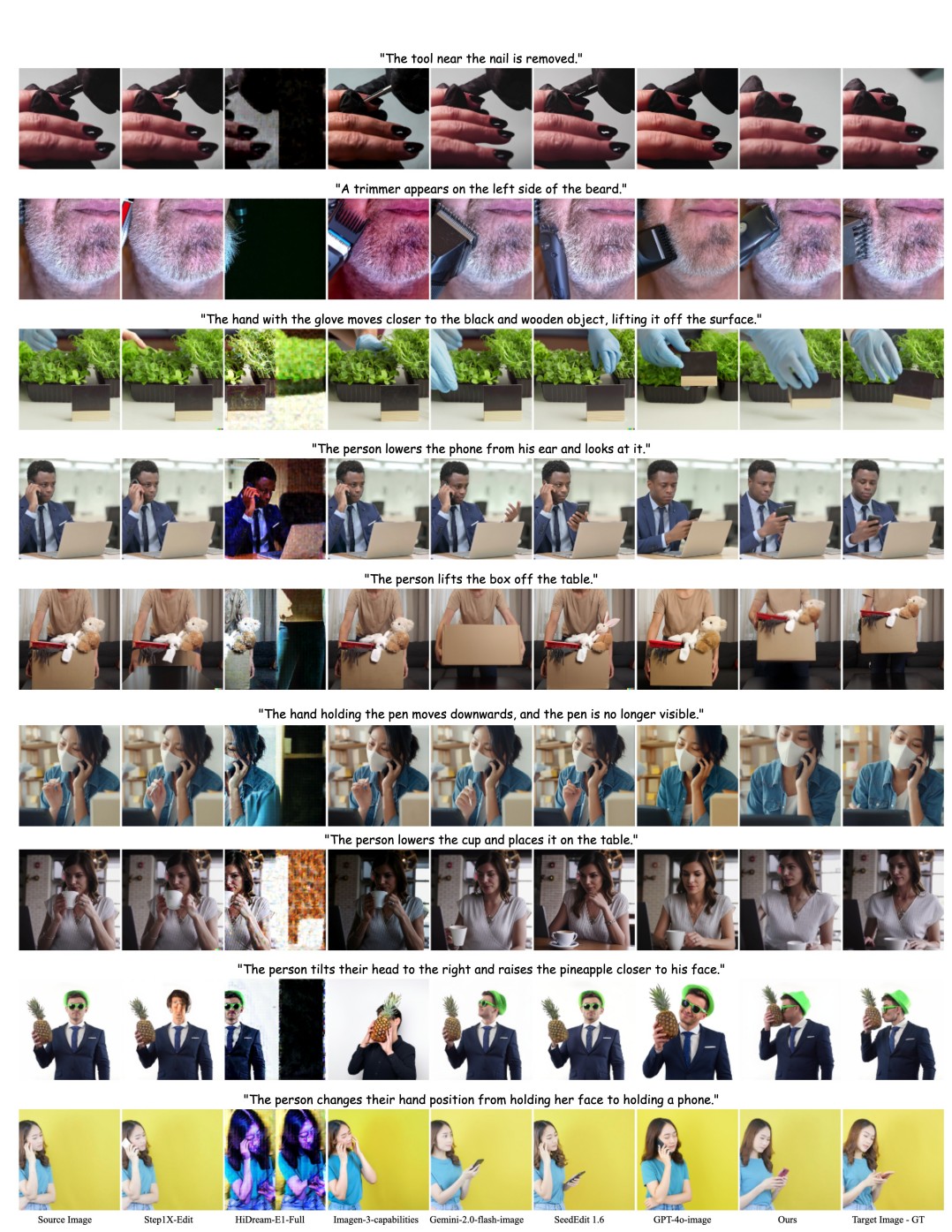

Figure 11: **Visualized comparison between methods from the industry on ByteMorph-Bench.** Editing Category: Human-Object Interaction.

