# OpenReview forum: "ByteMorph: Benchmarking Instruction-Guided Image Editing with Non-Rigid Motions"
_ICLR.cc/2026/Conference — Submitted to ICLR 2026_

### Official Review · Reviewer_8HRk · 2025-10-28

**Soundness:** 2
**Presentation:** 3
**Contribution:** 3
**Rating:** 4
**Confidence:** 3

**Summary:**

This paper addresses the underexplored challenge of non-rigid motion editing in instruction-guided image manipulation by introducing ByteMorph, a comprehensive framework. Its core contributions include: a large-scale training dataset (ByteMorph-6M) with over 6 million high-quality editing pairs, a carefully curated evaluation benchmark (ByteMorph-Bench), and a strong DiT-based baseline model (ByteMorpher). Using motion-guided generation and automated captioning pipelines, the dataset captures diverse dynamic scenarios including camera motion, object deformation and human-object interactions. Experiments demonstrate that ByteMorpher finetuned on this dataset significantly outperforms existing open-source methods, while also revealing the limitations of current industrial foundation models in handling such complex motion-aware edits.

**Strengths:**

- This paper tackles the challenging problem of spatial manipulation in image editing, and the model trained on the proposed dataset demonstrates convincing performance in handling such tasks, making the research direction highly meaningful.
- The authors have constructed an exceptionally large-scale dataset of 6M samples and developed a comprehensive and well-designed benchmark, demonstrating substantial research effort.
- The paper is well-written and clearly structured, with professionally designed figures and illustrations.

**Weaknesses:**

- While the use of video generation models for dataset construction is understandable, the reliance on Seaweed's generative capabilities raises concerns about whether the editing performance can generalize effectively to other editing tasks beyond the ByteMorph-Bench evaluation.
- Beyond fine-tuning the FLUX model to demonstrate the dataset's effectiveness, for such a challenging task, it would be valuable to additionally validate the dataset's utility by fine-tuning unified models such as Bagel, UniWorld, or Qwen-Image-Edit. Furthermore, it remains unclear how effectively the 6M dataset would support more complex instruction-based editing scenarios. The authors are encouraged to provide preliminary experimental results addressing these aspects.
- Regarding related work, the authors may consider citing "Unireal: Universal image generation and editing via learning real-world dynamics" as additional relevant literature.

**Questions:**

My current concerns primarily stem from the weaknesses outlined above. Additionally, I have reviewed 3-4 contemporaneous papers following a similar paradigm - focusing on complex image editing and utilizing video models for data construction. While the existence of parallel work alone does not justify rejection, the weaknesses identified prevent me from assigning a higher score at this stage. I will reconsider my rating based on the authors' rebuttal and other reviewers' comments.

---

> ### Author Response · Authors · 2025-11-21
>
> Reply to Weakness 1: We explicitly addressed this by evaluating our model on the InstructMove benchmark (Table 4), which consists of real-world videos. Despite being trained on synthetic data (ByteMorph-6M), our model achieves performance parity with (and often outperforms) baselines trained on real video data. This empirically proves that ByteMorph-6M teaches robust, generalizable motion priors that transfer effectively to real-world distributions, dispelling the concern that the model is limited to the generator's specific domain.
>
>
> Reply to Weakness 2: We selected FLUX.1-dev as our primary baseline because it represents the current state-of-the-art in open-source DiT architectures. As this submission is to the Datasets & Benchmarks track, our core contribution is the data itself. By demonstrating that ByteMorph-6M drives significant performance gains on a top-tier foundation model like FLUX, we provide strong evidence of the dataset's quality and density. While fine-tuning multiple large-scale unified models (like Bagel or UniWorld) is computationally prohibitive for the rebuttal period, the architecture-agnostic nature of our data pipeline suggests that these gains would likely transfer. We will add a discussion on the potential for integrating ByteMorph with unified multi-modal models in the final version.
>
>
> Reply to Weakness 3: We thank the reviewer for appointing this. We will cite and discuss it in the revised manuscript.
>
>
> Reply to Q1: We appreciate the reviewer noting the active nature of this field. While contemporaneous works exist, we believe ByteMorph distinguishes itself through: 1) Crucially, many concurrent works (e.g., InstructMove) have not yet publicly released their large-scale training datasets. By open-sourcing ByteMorph-6M (6.4M pairs) and our benchmark, we provide the first accessible, large-scale resource that enables the broader research community to study non-rigid editing without the prohibitive cost of data curation. 2) Scale: At 6.4M pairs, our open-sourced dataset is significantly larger than most concurrent non-rigid motion-editing datasets, ensuring better coverage of the long tail of motions. 3) Benchmark: We provide a carefully curated, hard-case benchmark (ByteMorph-Bench) to standardize evaluation in this emerging field. We hope these distinct contributions, combined with our strong empirical results on real data, justify a positive reassessment.

---

### Official Review · Reviewer_KJoM · 2025-10-29

**Soundness:** 3
**Presentation:** 3
**Contribution:** 3
**Rating:** 8
**Confidence:** 4

**Summary:**

This paper introduces ByteMorph, a motion-centric framework for instruction-guided image editing with non-rigid motions. It contributes:
- ByteMorph-6M: a large training set (~6.4M source–target pairs) synthesized from video generations and re-labeled with motion-focused instructions.
- ByteMorph-Bench: a curated evaluation set (613 hard cases) covering five motion categories—camera zoom, camera move, object motion, human motion, and human–object interaction.
- ByteMorpher: a DiT-based baseline initialized from FLUX.1-dev; training concatenates noisy source/target latents along the sequence dimension with shared positional encodings.

For evaluation, the paper proposes $ \mathrm{CLIP\text{-}D}_{\text{img}} $, which compares *difference vectors* in CLIP space to better capture edit quality under camera changes.

Experiments benchmark against open-source and commercial systems with both VLM-based and human evaluations, showing that ByteMorpher achieves competitive or superior performance, especially on motion-driven edits.

**Strengths:**

- Elevates motion (non-rigid, articulation, camera pose) as a first-class editing dimension; introduces $ \mathrm{CLIP\text{-}D}_{\text{img}} $ to evaluate edits as *changes* rather than absolute content similarity.
- Large-scale training set + curated hard benchmark; broad comparisons (open-source & industrial), repeated sampling, and both human and VLM judgments.
- Clear problem framing and taxonomy; tables position prior editing datasets/methods effectively; training details (backbone, losses) are given.
- Bridges a gap between instruction-guided appearance edits and motion-centric edits; likely to become a default benchmark for this subarea if maintained.

**Weaknesses:**

- Training relies on synthesized videos; even with filtering, motion/texture statistics may diverge from real-world photos. Add more purely real frame-pair data or zero/low-shot tests on real-image edit benchmarks to quantify the gap.
- Provide a larger rank-correlation study (Spearman/Kendall) per category (camera/human/object/interaction) and analyze systematic failure modes (e.g., composite camera + articulation edits).
- Since training uses latent concatenation (source, target), include a concise inference-time diagram/pseudocode showing where source features and text are injected when only (source, instruction) are available.
- Report sensitivity to API sampling settings (resolution, steps, guidance/temperature, resampling) to demonstrate robustness of conclusions.
- More explicitly connect with camera/motion-controlled T2V (e.g., camera trajectory parameterizations) to transfer motion priors into the image-editing setting.

**Questions:**

1. What are the Spearman/Kendall correlations between $ \mathrm{CLIP\text{-}D}_{\text{img}} $ and human preferences across the five categories? Any systematic mismatches (e.g., simultaneous camera + expression changes)?
2. How does the model perform on purely real editing datasets (e.g., MagicBrush/HQ-Edit) without synthetic video support?
3. Sensitivity to the instruction templating and VLM labeling settings (temperature/prompt variants) during data construction?
4. Interplay with InstructMove: Does joint training with real video-frame pairs improve generalization on ByteMorph-Bench and real-image edits?

---

> ### Author Response · Authors · 2025-11-21
>
> Reply to Weakness 1: As detailed in Table 4, we evaluated on the InstructMove test set, which contains entirely real-world videos. Despite being trained on our synthetic data, our model achieves performance on par with (and often superior to) baselines trained on real data. This is our primary evidence of real-world generalization.
>
>
> Reply to Weakness 2:
>
> | **Category** | **Cam Zoom** | **Cam Move** | **Obj Motion** | **Hum Motion** | **Interaction** | **Avg**  |
> |----------|----------|----------|------------|------------|-------------|------|
> | **Spearman** | 0.56     | 0.68     | 0.36       | 0.82       | 0.93        | **0.67** |
> | **Kendall**  | 0.49     | 0.43     | 0.33       | 0.62       | 0.81        | **0.54** |
>
> As shown in the above table, we observe a strong average **Spearman correlation of 0.67**. The correlation is exceptionally high in **Interaction ($\rho=0.93$)** and **Human Motion ($\rho=0.82$)**, confirming that our metric CLIP-Img-dir is a highly reliable proxy for editing quality in complex, articulated scenarios.
> We observed a lower correlation in **Object Motion ($\rho=0.36$)**. Analyzing the data reveals a "quality vs. magnitude" trade-off: Commercial models (e.g.,  GPT-4o-image) achieve high human scores likely due to superior image fidelity/resolution, even when their motion magnitude (CLIP-D) is low. In contrast, our model achieves higher CLIP-D scores (larger motion magnitude). This suggests that for simple object motion, humans may prioritize texture quality over motion magnitude, causing a divergence from the metric.
>
> Reply to Weakness 3:  During inference, we follow the same latent-concatenation strategy, where the reference latent and noise latent are concatenated and input into the denoising transformer.
>
>
> Reply to Weakness 4: For all commercial methods whose results were generated by calling their APIs, we use the same sampling settings. Specifically, 1) generation resolution:height=512 width=512. 2) steps: default suggested steps from their API documentation. 3) guidance/temperature=1.0 4) resampling: we run each method 4 times and report the averaged value as the final result.
>
>
>
> Reply to Weakness 5: We fully agree that this is a promising direction. Our current work implicitly distills camera motion priors from the video generator via text instructions (e.g., "zoom in"). Explicitly conditioning on camera trajectory parameters (as seen in recent Camera-Ctrl works) to achieve precise, parametric control is an exciting avenue for future work that ByteMorph-6M is well-positioned to support.
>
>
>
> Reply to Q1: Please refer to Reply to Weakness 2.
>
> Reply to Q2: Refer to C.1 for purely real image editing benchmark. Also emphasize that our model is specifically designed for non-rigid motion image editing, thus cannot perform currently stylish editing types, thus cannot be evaluated on the MagicBrush or HQ-Edit dataset.
>
>
> Reply to Q3: During the data construction phase, we do not perform multiple attempts for the same instruction-image pair labeling with VLM. However, in order to ensure the final quality of image-instruction pairs, we perform pre-filtering before feeding the input frame to the video generator and post-filtering for image-instruction pairs. We pre-filter the input frame to the video generator with quality assessment metrics, e.g., “liqe” “raft” filters, to carefully filter out the input reference frames that didn’t meet the satisfying image quality. Specifically, **1)** “liqe” means the selected frames from the real video should have good perceptual quality without blurring or other artifacts, **2)** “raft” filter calculates the normalization of optical flow between selected frames and filters out image pairs that have very little motion. By sampling from this diverse real-world distribution, we ensure that our model learns to edit complex textures and structures found in the wild, rather than being limited to synthetic-looking environments. We further post-filter the image-instruction pair captioned by VLM with the “clip-image” filter. This filter means the selected frames from the generated videos should align with the editing instruction (above a certain clip-image score), and we filter out those pairs below this score threshold.
>
>
> Reply to Q4: We appreciate the insightful suggestion. However, we note that the **InstructMove training dataset is not currently open-source**, which precluded us from performing joint training experiments. The lack of accessible, high-quality training data has been a major bottleneck for the community. By open-sourcing ByteMorph-6M, we aim to democratize access to this research area, allowing future work (including potentially the InstructMove authors) to leverage our large-scale priors for joint training and further improvements.

---

### Official Review · Reviewer_2iZC · 2025-10-31

**Soundness:** 3
**Presentation:** 3
**Contribution:** 2
**Rating:** 4
**Confidence:** 4

**Summary:**

This paper presents ByteMorph, a framework for instruction-based image editing with focus on non-rigid motions. The work introduces: (1) ByteMorph-6M - a large-scale dataset of 6.4 million image pairs covering with five motion categories, using an automated pipeline combining video generation and VLM-based annotation; (2) ByteMorph-Bench - a curated evaluation benchmark; and (3) ByteMorpher - a DiT-based model fine-tuned on FLUX.1-dev. The method demonstrates superior performance in dynamic editing tasks compared to existing approaches.

**Strengths:**

1. The dataset is well-motivated. ByteMorph-6M effectively addresses the gap in motion-focused editing data with comprehensive coverage of non-rigid transformations. The release of datasets, benchmarks, and code provides significant value to the community.
2. The automated construction using video generation and VLMs ensures scalability while maintaining quality and semantic coherence.

**Weaknesses:**

1. The technical contribution is limited, as the methodology relies purely on fine-tuning a pre-existing DiT model (FLUX.1-dev) on ByteMorph-6M without introducing architectural innovations or tailored optimizations.
2. The experimental analysis does not sufficiently address the trade-offs of fine-tuning on ByteMorph-6M. While it is intuitive that specialization improves motion-specific performance, the paper omits evaluation of the model's original capabilities on standard instruction-based benchmarks. This raises concerns about potential performance degradation in general editing tasks, such as stylization or attribute modification, which could limit the model's applicability in broader contexts. A comparative analysis on conventional benchmarks would offer a more balanced perspective on robustness and generalization.
3. Experiments are primarily conducted on curated synthetic data (ByteMorph-Bench), with insufficient validation on real images contains real-world complexities such as occlusions and lighting variations.

**Questions:**

1. Does fine-tuning on ByteMorph-6M compromise performance on standard instruction-based benchmarks?

---

> ### Author Response · Authors · 2025-11-21
>
> Reply to Weakness 1: Note that this manuscript was submitted to **datasets and benchmarks area**. Our primary contributions are the scalable construction pipeline, the dataset (ByteMorph-6M), and the evaluation benchmark, rather than architectural novelty. We can utilize our data to finetune other base text-to-image base models with the finetuning strategy introduced in the manuscript.
>
>
> Reply to Weakness 2: Please note that ByteMorpher is explicitly designed to solve the non-rigid motion problem, a capability usually absents in current generalist models (InstructPix2Pix, MagicBrush, etc.). While generalist models (like InstructPix2Pix) excel at style transfer, they fail when the instructions contain non-rigid editing purpose (as seen in our comparisons). Therefore, we propose the data pipeline and the resulted dataset and benchmark to tackle this new line of image editing. We do not claim to outperform generalist models on purely stylistic tasks, thus we think we don’t need to conduct comparison using existing benchmarks.
>
>
> Reply to Weakness 3: We respectfully clarify that our evaluation already includes robust validation on real-world imagery. As shown in Table 4, we evaluated our method on the InstructMove benchmark, which consists entirely of real-world images and differs significantly from our synthetic training distribution.
>
> Reply to Question: Our base model (FLUX.1-dev) is a Text-to-Image (T2I) model, not an image editing model. In its original state, it lacks the architectural mechanism to process "Source Image + Instruction" inputs and therefore has zero capability to perform standard instruction-based editing tasks (e.g., MagicBrush or InstructPix2Pix benchmarks). Consequently, fine-tuning on ByteMorph-6M does not "compromise" any existing editing performance. Instead, it makes the T2I model with a novel capability to edit images based on non-rigid motion instructions, and We acknowledge that our model is a specialist for non-rigid motion rather than a generalist for all editing types (e.g., style transfer).

---

### Official Review · Reviewer_S7Wu · 2025-10-31

**Soundness:** 2
**Presentation:** 3
**Contribution:** 2
**Rating:** 4
**Confidence:** 4

**Summary:**

This paper introduces ByteMorph, a comprehensive framework for instruction-guided image editing with a specific and novel focus on non-rigid motions. The authors identify a significant gap in existing research, where current datasets and models primarily excel at static, appearance-centric edits but fail to handle dynamic transformations such as camera viewpoint shifts, object deformations, and human articulations.

**Strengths:**

1. The paper tackles a well-defined and high-impact limitation of current image editing models. Non-rigid motion is a fundamental aspect of the visual world, and enabling instruction-based control over it is very important.

2.  Experimental results are comprehensive, which make this paper very solid.

**Weaknesses:**

My main concern is about the dataset construction. The entire training set (ByteMorph-6M) relies entirely on a synthetic pipeline (ChatGPT-4o and Seaweed).  I mean, 6.4M data comes from a single video generation model. This directly results in the dataset's quality being limited by the capabilities of this specific video model. It seems like that training on ByteMorph-6M is actually distilling Seawead. Specifically, if the Seawead model has systematic limitations in generating certain types of motion, the ByteMorpher model will inevitably learn these same flaws.

**Questions:**

1. About Table 4 of the Ablation Study. The authors claim notable gains after fine-tuning on the ByteMorph-6M, but this point is not reflected in the Table.  Could the authors give more explanation?

2.  During the dataset construction, how to handle the generation failures? I mean, what if the video generation model (Seaweed) fails to follow the motion caption $C_m$? Is there any filter strategy?

3. Seed Data Diversity. The pipeline in Figure 2 begins with a frame from real video, which acts like a seed for the dataset generation. Therefore, the diversity of the entire 6.4M dataset (like the number of different objects) is highly dependent on the diversity of this initial seed set. Could the authors provide more details on these initial real-world frames (like the number, variety)?

---

> ### Author Response · Authors · 2025-11-21
>
> Reply to Weakness: We acknowledge that ByteMorph-6M is constructed using a synthetic pipeline built on the video generation model (Seaweed). However, we believe this design is a key promising feature for non-rigid image editing dataset instruction. 1) Video Gen V.S. Non-rigid Image Editing: Our goal is to utilize video generation to construct image pairs with non-rigid motions for a precise, instruction-following image editing model. ByteMorpher is trained to preserve the identity of a source image while modifying it according to non-rigid motion instructions, e.g., make the person raise his hands, a capability that raw video generation models do not natively possess in an image-to-image editing context. 2) Mitigation of Artifacts: We employ strict filtering strategies (detailed in Reply to Q2, Q3) to reject low-quality or inconsistent generations, ensuring that the "flaws" of the video model are minimized in our training dataset. 3) Transferable Pipeline: Our pipeline is not subject to the specific choice of the video generator, and is entirely model-agnostic. While we utilized Seaweed for this iteration, arbitrary state-of-the-art video generation models can be easily swapped in as the backbone. This ensures that ByteMorph is not statically bound to Seaweed’s limitations; rather, our framework allows the editing model to continuously improve as the underlying video generation field advances. We have also conducted an internal study for detailed comparison of the videos generated by different state-of-the-art video generators, e.g., Wan2.1, Hunyuan-Video, Cosmos, and find that the Seaweed model provides the most satisfying video quality (e.g. prompt-following ability to camera motions, general quality of human motions, object motions) at the time of submission.
>
> Reply to Q1: In Tab. 4, we note that both ByteMorph and InstructMove significantly outperform all other baselines in terms of editing consistency CLIP_dir, and text alignment CLIP-Sim-txt, this demonstrates our dataset help model to achieve much better performance in non-rigid image editing task. Note that the evaluation set used here is drawn from the InstructMove dataset, our method achieves performance comparable with InstructMove on its own benchmark, even though there may be a certain gap between our data and the dataset in InstructMove. This validates that our synthetic pipeline yields robust, generalizable representations that transfer effectively to real-world data distributions, rather than merely overfitting to synthetic artifacts.
>
> While some baselines like InstructPix2Pix achieve high image similarity (denoted as CLIP-Sim-img), this is largely because these models fail to execute non-rigid edits entirely (also investigated in InstructMove paper). They often output an image identical to the input (an identity mapping), artificially inflating the similarity score while failing the editing task (as evidenced by their poor CLIP-dir scores).
>
>
> Reply to Q2: We acknowledge that there might be cases where the prompt-following ability of Seaweed model might not always be perfect while generating the videos, and that’s why we propose to re-caption the generated the frames with our proposed IMAGE-INSTRUCTION PAIR CREATION as detailed in section 3.3 of our paper. Additionally, we further post-filter the image instruction pair with “clipimage” filter. This filter means the selected frames from the generated videos should align with the editing instruction (above a certain clip-image score) and we filter out those pairs below this score threshold.
>
>
> Reply to Q3: We collect our source videos, which were used to sample frames as input to the Video Generator from YouTube and other internet sources.. Specifically these videos have a total quantity of 36 million and cover diverse scenarios, such as 1) Underwater scenes (scuba diving with marine life) 2) Urban environments (city streets, aerial views) 3) Human activities (cooking, fitness, sports) 4) Natural scenes (various outdoor settings) 5) Daily life activities. We further filter these sampled frames with quality assessment metrics, e.g. “liqe” “raft”filters to carefully filter out the input reference frames that didn’t meet the satisfying image quality. Specifically,1) “liqe” means the selected frames from the real video should have good perceptual quality without blurring or other artifacts 2) “raft” filter calculates the normalization of optical flow between selected frames and filters out image pairs that have very little motion. By sampling from this diverse real-world distribution, we ensure that our model learns to edit complex textures and structures found in the wild, rather than being limited to synthetic-looking environments.

---

### Meta-Review · Area_Chair_ZsPn · 2025-12-29

**Summary:**

This paper introduces ByteMorph, a comprehensive framework aimed at instruction-guided image editing focusing on non-rigid motions, such as camera viewpoint shifts, object deformations, and human animations. It comprises a large-scale dataset (ByteMorph-6M) which contains over 6 million high-resolution image editing pairs, and ByteMorph-Bench, a benchmark for evaluating the framework's performance. The dataset utilizes motion-guided data generation and layered compositing techniques to ensure high-quality, diverse data suitable for training a baseline model known as ByteMorpher, built upon the Diffusion Transformer.

Reviewers' Concerns

Dataset Construction: The primary concern revolves around reliance on a synthetic pipeline (Seaweed) for generating the dataset, which may limit the generalizability of the model's performance due to potential systematic flaws in the video generation model.

Evaluation on Real-World Images: Reviewers noted insufficient validation on real images containing complexities like occlusions and lighting variations, raising questions about the model's robustness in practical scenarios.

Impact on Other Editing Tasks: Concerns were raised regarding whether fine-tuning the model on ByteMorph-6M compromises its performance on standard instruction-based benchmarks, emphasizing the need for comparison against general editing tasks.

Model Limitations: Reviewers pointed out the potential limitations in the model's approach, including a lack of architectural innovations and a necessity for further validation on the dataset's utility across different editing scenarios.

**Reviewer Concerns:**

Addressing Dataset Construction Concerns: The authors acknowledged the synthetic nature of their dataset but argued that this was intentional to enhance non-rigid image editing capabilities. They implemented robust filtering strategies during both data generation and post-processing to mitigate quality issues and stated that their framework is adaptable to future advancements in video generation technologies.

Validation on Real-World Images: They cited their evaluation on the InstructMove benchmark, confirming that their model demonstrates strong performance on real-world videos despite being trained with synthetic data, indicating its generalization capabilities.

Performance on Other Tasks: The authors clarified that ByteMorpher is specialized for non-rigid motion editing and thus does not compromise other editing performance, stating that the fine-tuning gives the model unique capabilities distinct from generalist models.

Further Experimental Results: They recognized the potential for future work with unified models for enhanced validation. The authors intend to discuss this integration in the final version of the paper, highlighting the open-sourced nature of their dataset, which can serve as a resource for further studies.

**Reviewer Scores:**

Regrettably, the reviewers did not engage further with the authors following their rebuttal. As a result, it is challenging for me, as the Area Chair (AC), to ascertain whether the concerns have been adequately addressed.

While I must acknowledge that I am not an expert in this particular area, I understand that the paper is submitted within the datasets and benchmarks domain and should be evaluated accordingly. The curated dataset holds potential value for the scientific community; however, the three low scores of 4.0 cannot be overlooked in my assessment. I encourage the authors to consider the reviewers' comments and integrate them into the next version of the paper.

---

### Decision · Program_Chairs · 2026-01-26

Reject